# Towards Expanding-Node Spatial-Temporal Forecasting: A Structured Node Interaction Prompting Perspective

## Abstract

The rapid expansion of sensor systems, such as traffic networks, climate monitoring, and energy scheduling, poses new challenges for spatial-temporal series forecasting. While existing models have achieved strong performance under the fixed-node assumption, they rely on node-dependent parameters and fail to adapt when the network evolves, i.e., when old nodes are removed and new nodes with limited history are added. This expanding-node forecasting scenario introduces two critical challenges: (1) learning heterogeneous node representations without coupling learnable parameters to node count, and (2) enabling effective adaptation to new nodes with scarce observations. To tackle these challenges, we propose SNIP (Structured Node Interaction Prompting), a model-agnostic framework that constructs static spatial-temporal priors from historical observations and topology, and dynamically refines them during model training. Specifically, SNIP generates structured priors from three perspectives: periodic patterns across nodes, spatial-temporal interactions under time delays and graph structural information. These priors are projected into model as node promptings and then dynamically refined. For new nodes, SNIP initializes priors by similarity-weighted mixtures of old nodes and updates them with limited history, enabling efficient few-shot adaptation. Extensive experiments on multiple datasets demonstrate that SNIP outperforms state-of-the-art baselines in expanding-node scenarios. Beyond accuracy, SNIP provides plug-and-play generality and computational efficiency, bridging the gap between fixed-node precision and expanding-node adaptability in spatial-temporal forecasting.

## 1 Introduction

Spatial-temporal forecasting is crucial in cyber-physical systems such as traffic networks, climate monitoring, and energy scheduling. Despite recent advances, most models still rely on the ***fixed-node assumption***: training and inference are performed on a static node set, with parameters explicitly tied to node count. However, real systems are rarely static. Nodes may be *added* (e.g., new road sensors, weather stations) or *removed* (e.g., failures, replacements). This gives rise to the task of ***expanding-node forecasting***, where node sets evolve across periods, new nodes have scarce history, and some old nodes disappear, rendering traditional models ineffective.

To address this challenge, three lines of solutions have emerged (Figure 1): (1) **Node-independent parameterization.** Univariate time-series forecasting models forecast each node separately, which is scalable but neglects cross-variable dependencies. Others remove node embeddings and rely solely on sequence interactions (Liu et al., 2024; Ma et al., 2025a), while attention-based prompting (Hu et al., 2024) alleviates this partially but remains constrained by short horizons. (2) **Node-scaled Prompting.** Continual learning methods expand embeddings as new nodes appear (Chen & Liang, 2025), but usually assume abundant expansion data, which is unrealistic under scarcity. They also overlook node removal, causing wasted parameters and reduced flexibility. (3) **Fixed expanded parameterization.** A recent work, STEV (Ma et al., 2025b), introduces the Expanding-variate Time Series (EVTSF) forecasting task and mitigates imbalance with a flat scheme and shared subgraph. Nonetheless, it still relies on predefined embeddings for all expanded nodes. As a result, further network changes require costly retraining, limiting scalability.

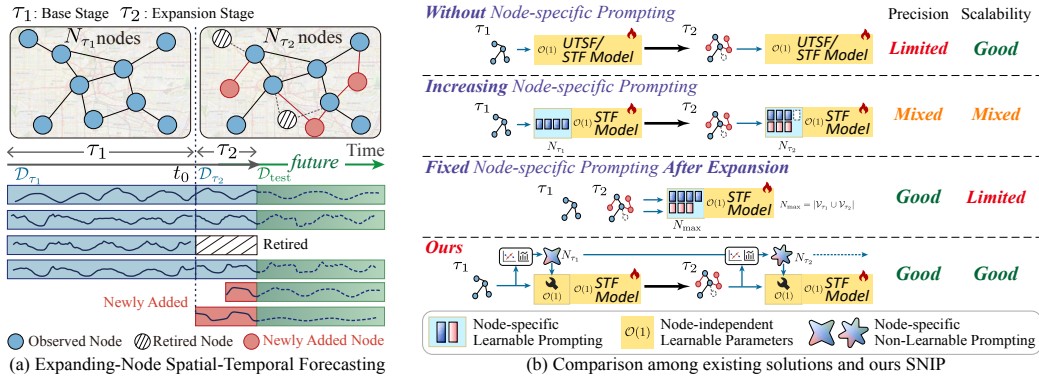

Figure 1: Examples of expanding-node spatial-temporal forecasting and different solutions. (a) Sensor nodes may added, retired or replaced in the expansion stage. (b) Comparison of three existing solution paradigms with our proposed SNIP framework.

In summary, while node-specific learnable parameters enhance forecasting accuracy (Shao et al., 2022; Liu et al., 2023; Dong et al., 2024), they either lack flexibility for new nodes when fixed or suffer from poor fitting under data scarcity when expanded, leading to a trade-off between accuracy and scalability. As a result, this raises a fundamental question:

*Can effective node identification features be computed directly from historical observations, without relying on learnable node-dependent model parameters?*

However, two critical challenges emerge: *(Challenge 1) How to ensure that constructed features sufficiently capture inter-node heterogeneity and correlation*, preserving predictive accuracy comparable to learnable embeddings? *(Challenge 2) How to refine these features dynamically to remain accurate under dynamic enviroments*, especially when new nodes arrive with only scarce observations?

To address these challenges, we propose SNIP (Structured Node Interaction Prompting), a model-agnostic prompting framework guided by structured priors and refined dynamically. Specifically, SNIP addresses the first challenge by computing priors from historical sequences through dimensionality reduction, which inherently preserves heterogeneity and correlation. Using PCA-based periodic features and Spectral embeddings of time-delayed interactions and graph topology, it effectively encodes node-specific heterogeneity without learnable embeddings. To tackle the second challenge, SNIP incorporates a dynamic refinement module that continuously adapts static priors through diffusion-based graph convolutions, thereby maintaining accuracy under dynamic evolving. Moreover, for new nodes with scarce observations, SNIP introduces a similarity-weighted initialization scheme that transfers priors from old nodes, providing effective embeddings that enable rapid few-shot adaptation. Through these two strategies, SNIP achieves parameter-node decoupling while maintaining both predictive accuracy and adaptability in expanding-node forecasting. Our contributions can be summarized as follows:

- We identify the problem of expanding-node spatial-temporal forecasting, where sensor networks evolve across periods, and highlight its core challenges of parameter-node coupling, data scarcity for new nodes, and preserving node heterogeneity. We further approach this problem from the perspective of structured node interactions.

- We propose SNIP (Structured Node Interaction Prompting), a framework that combines static prior construction (periodic, topological and time-delayed node interaction features) with dynamic refinement to build effective and flexible node promptings. In addition, we design a similarity-weighted initialization scheme to endow new nodes with initial embeddings, enabling efficient adaptation under few-shot conditions.

- A concrete instantiation of SNIP, termed SNIPformer, is further proposed. Extensive experiments on four datasets demonstrate that SNIP outperforms state-of-the-art baselines. Moreover, it serves

as a plug-and-play module that enables classical spatial-temporal models to adapt flexibly and effectively to expanding-node forecasting.

## 2 RELATED WORK

Spatial-temporal forecasting (STF) is central to applications such as traffic, energy, and climate. Early works combined recurrent or convolutional networks with graph modules to model temporal and spatial dependencies. With the advent of Spatio-Temporal Graph Neural Networks (STGNNs) and Transformers (Li et al., 2018; Wu et al., 2019; Guo et al., 2022), research has focused on capturing complex inter-node correlations via multi-view graphs or attention (Diao et al., 2024; Jiang et al., 2023). More recent advances explore adaptive embeddings (Shao et al., 2022; Zheng et al., 2025a) and hybrid neural modules (Sun et al., 2024; Lee & Ko, 2024) to balance efficiency and accuracy.

**Node Prompting in STF.** A consistent trend in these developments is the introduction of node-specific embeddings as additional identity information. By assigning learnable parameters to each node, models can capture inter-node heterogeneity beyond raw time series, which has shown strong forecasting performance (Liu et al., 2023; Dong et al., 2024). Such embeddings function as prompts that guide spatio-temporal modules, and have become an implicit consensus for achieving state-of-the-art accuracy. However, this design inherently ties model parameters to node count, limiting scalability in evolving networks. Recent work has further explored attention-based prompting mechanisms, such as STGP (Hu et al., 2024) and EAC (Chen & Liang, 2025), but these methods still rely on directly fitting prompts from data, which is challenging and assumes the availability of sufficient training samples.

**STF under dynamic node expansion.** In real-world systems, nodes may be added or removed over time, violating the fixed-node assumption in classical STF. To address this, recent works explored several directions. One approach decomposes data into univariate series or removes node-specific embeddings, which improves scalability but ignores spatial dependencies. In addition, node-count-agnostic models like literature (Altieri et al., 2024; Li et al., 2018; Zheng et al., 2020) can also handle changing node sets, but they mainly rely on short-window inputs and ignore node-specific inherent heterogeneity. Others directly learn from raw inputs or attention-based prompts (Liu et al., 2024; Hu et al., 2024), but accuracy drops due to insufficient heterogeneity modeling. Continual learning methods (Wang et al., 2023; Chen & Liang, 2025) expand embedding sets through prompt-tuning, yet typically assume abundant new data. OOD-generalization based methods (Wang et al., 2024; Ma et al., 2025a) emphasize robustness but lose accuracy when fine-tuning is feasible. A recent EVTSF paradigm, STEV (Ma et al., 2025b), mitigates imbalance via flattening and contrastive learning, but still relies on node-dependent parameters and costly retraining, limiting flexibility.

In contrast, our SNIP builds non-learnable priors and refines them dynamically, decoupling parameters from nodes while retaining node-specific effectiveness, and can be seamlessly integrated into existing STF models.

## 3 PRELIMINARY

We consider a spatial-temporal network at time period $\tau$, denoted as $\mathcal{G}_\tau = (\mathcal{V}_\tau, \mathcal{E}_\tau)$, where $\mathcal{V}_\tau = \{v_1, v_2, ..., v_{N_\tau}\}$ is the node set (e.g., road sensors, climate monitors), and $\mathcal{E}_\tau$ denotes the edges (e.g., road links, physical connections). The adjacency matrix is $A_\tau \in \mathbb{R}^{N_\tau \times N_\tau}$, $N_\tau = |\mathcal{V}_\tau|$, representing the spatial relationships among nodes. Each node records $C$ features within a temporal window of length $L$, forming a spatial-temporal series $\mathbf{X}_\tau \in \mathbb{R}^{L \times N_\tau \times C}$.

**Definition (Expanding-node Spatial-Temporal Series).** We define two consecutive periods. Period-1 (**base stage**) is $\tau_1 = [t_0 - L_1 + 1, t_0]$, with data $\mathcal{D}_{\tau_1} = (\mathcal{G}_{\tau_1}, \mathbf{X}_{\tau_1})$, where $|\mathcal{V}_{\tau_1}| = N_{\tau_1}$. $L_1$ denotes the length of base stage, and $t_0$ is the final time slice of this stage. Period-2 (**expansion stage**) is $\tau_2 = [t_0 + 1, t_0 + L_2]$, with data $\mathcal{D}_{\tau_2} = (\mathcal{G}_{\tau_2}, \mathbf{X}_{\tau_2})$, where $|\mathcal{V}_{\tau_2}| = N_{\tau_2}$. $L_2$ is the length of expansion stage. During the transition from the base stage to the expansion stage, nodes may be *added* or *removed*, which can be formalized as $\mathcal{V}_{\tau_2} = (\mathcal{V}_{\tau_1} \setminus \mathcal{V}_{del}) \cup \mathcal{V}_{new}$, $\mathcal{V}_{del} \subseteq \mathcal{V}_{\tau_1}$, $\mathcal{V}_{new} \cap \mathcal{V}_{\tau_1} = \emptyset$. Moreover, to enable timely forecasting on newly deployed nodes, the available data in the expansion

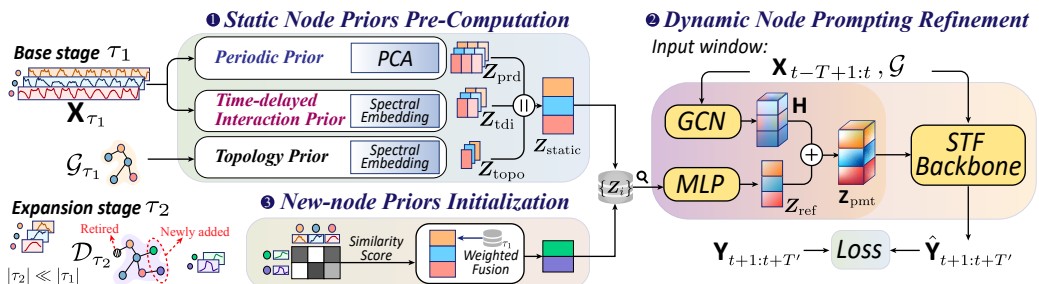

Figure 2: Overall framework of our proposed SNIP.

stage is typically very limited, i.e., $L_2 \ll L_1$, which leads the problem of data scarcity, particularly for newly added nodes.

**Problem (Expanding-node Spatial-Temporal Forecasting).** The goal of expanding-node spatial-temporal forecasting is to train a model $f$ using data from both periods, such that $f$ : $(\mathbf{X}_{t-T+1:t}, \mathcal{G}_\tau; \Theta) \mapsto \hat{\mathbf{Y}}_{t+1:t+T'}$, where $\mathbf{X}_{t-T+1:t} \in \mathbb{R}^{T \times N_\tau \times C}$ is the input sequence of length $T$, and $\hat{\mathbf{Y}}_{t+1:t+T'} \in \mathbb{R}^{T' \times N_\tau \times C}$ is the predicted sequence of length $T'$. A key requirement is that the parameter set $\Theta$ be decoupled from network size, i.e., $|\Theta| = \mathcal{O}(1)$, since parameter scaling with $N$ limits adaptability to evolving networks. This enables the model to generalize across varying node sets. In practice, we evaluate forecasting on the expanded set $\mathcal{V}_{\tau_2}$, while the formulation naturally extends to any node set with $N_\tau \geq N_{\tau_1}$, ensuring applicability to future expansions.

## 4 METHODOLOGY

Figure 2 illustrates the overall structure of the proposed SNIP framework. In this section, we present the construction of structured static priors and their refinement during training and expansion. Then we introduce SNIPformer, an instantiation built on an efficient spatial-temporal encoder.

### 4.1 STRUCTURED STATIC PRIORS (CHALLENGE 1)

In recent years, node-specific learnable embeddings have been widely used in spatio-temporal forecasting to provide discriminative identity information (Shao et al., 2022; Liu et al., 2023; Dong et al., 2024; Chen & Liang, 2025), achieving strong performance but conflicting with evolving node sets. To address this, SNIP avoids node-dependent parameters and instead pre-computes node-specific priors from historical data as prompting signals. We derive low-dimensional features that maximize inter-node variance to preserve heterogeneity.

#### 4.1.1 PERIODIC PRIORS

**Motivation.** Intuitively, a node's long-term sequence itself serves as its unique identifier, but directly using it is impractical due to dimensionality and noise. We instead apply dimensionality reduction to extract informative components. Given the strong periodicity of spatial-temporal data (e.g., daily or weekly cycles), we partition histories into repeated cycles, compress each into low-rank "snapshots," and average them to form a compact representation of node identity.

**Periodic Priors.** Given a historical period $\tau$ with length $L$, $\mathbf{X} \in \mathbb{R}^{L \times N \times C}$ denote the historical sequence for $N$ nodes. For clarity, we describe the single-feature case ($C = 1$) below, which naturally extends to multi-channel inputs by concatenation. We specify a set of cycle lengths $\{p_1, p_2, \ldots, p_n\}$ and partition each node's series into non-overlapping segments accordingly. For instance, when $p_j$ corresponds to one day, the sequence is divided into consecutive daily fragments. Each segment is normalized independently, and then projected into a low-dimensional representation using Principal Component Analysis (PCA) (Abdi & Williams, 2010). For each cycle length $p_j$, the node representations from all complete segments are averaged to yield a compact descriptor $\bar{\mathbf{Z}}^{(j)} \in \mathbb{R}^{N \times k_{\text{pca}}}$, $k_{\text{pca}}$ represents the value of a low dimensionality. Finally, we concatenate results across all $n$ cycle

lengths to obtain the periodic priors:

$$\boldsymbol{Z}_{\text{prd}} = \text{Concat}\left(\bar{\boldsymbol{Z}}^{(1)}, \bar{\boldsymbol{Z}}^{(2)}, \dots, \bar{\boldsymbol{Z}}^{(n)}\right) \in \mathbb{R}^{N \times (n \cdot k_{\text{pca}})}. \tag{1}$$

By the Eckart-Young-Mirsky theorem (Eckart & Young, 1936), PCA guarantees the optimal rank-$k$ approximation under the Frobenius norm, thereby preserving the maximum variance. In our context, this ensures that the periodic priors retain the most discriminative directions of node dynamics, effectively encoding node ***heterogeneity*** from a temporal perspective. Implementation details are provided in Appendix A.1.

### 4.1.2 Topology Priors and Time-delayed Interaction Priors

**Motivation.** While periodic features separate node-specific temporal patterns, they overlook inter-node correlations, another key factor in spatial-temporal forecasting (Wang et al., 2022). Topology priors, derived from graph adjacency, capture latent positional and structural relations. Meanwhile, many spatial-temporal phenomena propagate with delays (e.g., traffic congestion spreading) (Long et al., 2024; Zheng et al., 2025a), and such delayed or short-term correlations are inherently dynamic. To capture these correlations, we construct two complementary priors: (1) topology features from static adjacency, and (2) time-delayed interaction features from frequency-domain correlations under short windows.

**Topology Priors.** We adopt spectral embedding (Belkin & Niyogi, 2003) to obtain low-dimensional node representations. In the case of topology, given the adjacency matrix $\boldsymbol{A} \in \mathbb{R}^{N \times N}$, one can construct the normalized Laplacian: $\boldsymbol{L} = \boldsymbol{I} - \boldsymbol{D}^{-\frac{1}{2}} \boldsymbol{A} \boldsymbol{D}^{-\frac{1}{2}}$, where $\boldsymbol{D}$ is the diagonal degree matrix with $D_{i,i} = \sum_j A_{i,j}$. Then, the topology embedding is formed by the eigenvectors corresponding to the smallest $k_{\text{topo}}$ eigenvalues of $\boldsymbol{L}$. This can be formulated as:

$$\boldsymbol{Z}_{\text{topo}} = \Phi(\boldsymbol{A}, k_{\text{topo}}) = [\boldsymbol{u}_1, \boldsymbol{u}_2, \dots, \boldsymbol{u}_{k_{\text{topo}}}] \in \mathbb{R}^{N \times k_{\text{topo}}}, \tag{2}$$

where $\boldsymbol{u}_1, \dots, \boldsymbol{u}_{k_{\text{topo}}}$ are the leading eigenvectors of the normalized Laplacian. This embedding captures the static positional structure of nodes in the network, where nearby or strongly connected nodes are embedded closer together. Physically, they reflects both global communities and local connectivity patterns.

**Time-delayed Interaction Priors.** Recent studies have shown that correlations between node sequences often emerge more strongly when temporal delays are considered, rather than assuming synchronous dynamics (Long et al., 2024; Zheng et al., 2025a). To capture such effects, we estimate cross power spectral density (CSD) between node pairs using Welch's method with short sliding windows (Welch, 1967). This formulation enables us to measure correlations across all possible lags without pre-specifying a maximum delay in previous STF models. Consequently, we can obtain the cross-correlation matrices under different time delays: $\boldsymbol{R}(\delta)$. From this, we extract (i) the dominant delay $\Delta_{i,j} = \arg\max_\delta |R_{i,j}(\delta)|$, that maximizes correlation between nodes $i$ and $j$, and (ii) the corresponding correlation strength $P_{i,j} = \max_\delta |R_{i,j}(\delta)|$. These two matrices encode how information propagates with delays and how strongly nodes interact. Similarly, we apply spectral embedding to both matrices, and then concatenate results into $\boldsymbol{Z}_{\text{tdi}}$:

$$\boldsymbol{Z}_{\text{tdi}} = \text{Concat}(\Phi(\Delta, k_{\text{delay}}), \Phi(\boldsymbol{P}, k_{\text{corr}})) \in \mathbb{R}^{N \times (k_{\text{delay}} + k_{\text{corr}})}. \tag{3}$$

In summary, topological priors preserve static, position-driven relationships, while time-delayed embeddings capture dynamic propagation and short-term coupling. In particular, spectral embeddings emphasize the principal eigenvectors, which correspond to directions of maximum structural or interaction variance, this is analogous to PCA but under graph constraints. These priors reflect how nodes interact and differ within the network, boosting promptings from the ***correlation*** angle. Details of CSD method are provided in the Appendix A.2.

### 4.2 Dynamic Refinement and Adaptation (Challenge 2)

**Motivation.** The static priors in Section 4.1.2 capture invariant properties but cannot reflect temporal dynamics, such as evolving behaviors of existing nodes or the emergence of new nodes during expansion. To address this, we design a refinement-and-adaptation mechanism that treats above three priors as reference points subject to dynamic correction. To formalize this intuition, we propose the

following hypothesis, which conceptualizes how an optimal node prompting should be decomposed into stable and dynamic components.

**Hypothesis 1** (Decomposition of Optimal Node Prompting). *At any time $t$, there exists an optimal prompting configuration $\boldsymbol{z}_{i\star}^{(t)}$ for each node $i$, which maximizes predictive accuracy. This configuration can be decomposed as: $\boldsymbol{z}_{i\star}^{(t)} = \boldsymbol{q}_i + \boldsymbol{r}_i^{(t)}, \boldsymbol{r}_i^{(t)} = g(\boldsymbol{x}_j^{(t)}, j \in \mathbb{N}_{(i)})$, where $\boldsymbol{q}_i$ represents spatially intrinsic characteristics of node $i$ (time-invariant reference), $\boldsymbol{r}_i^{(t)}$ reflects spatial-temporal interaction effects that vary over time, $\mathbb{N}_{(i)}$ is the set of nodes that have a correlation relationship with node $i$, and $g$ is a transformation function.*

This decomposition allows $\boldsymbol{q}_i$ to represent slowly varying identity, with $\boldsymbol{r}_i^{(t)}$ capturing fast, context-dependent deviations. In our study, long horizon priors built via multi cycle PCA and spectral embeddings preserve between node variance and aim to capture inherent node properties in $\boldsymbol{q}_i$. The refinement $\boldsymbol{r}_i^{(t)}$ then adapts these identities to current conditions during training. Through this, the learnable model only needs to fit $\boldsymbol{r}_i^{(t)}$. Consequently, the hypothesis space is constrained, yielding reduced sample complexity and improved generalization under limited data (Vapnik, 1999). We will provide empirical ablations in Section 5.3 to support this assumption.

**Dynamic refinement via MLP and diffusion graph convolution.** We first project the concatenated static priors into the model dimension $d$ using a two-layer MLP: $\boldsymbol{Z}_{\text{ref}} = \text{MLP}([\boldsymbol{Z}_{\text{prd}}, \boldsymbol{Z}_{\text{topo}}, \boldsymbol{Z}_{\text{tdi}}]) \in \mathbb{R}^{N \times d}$. We refine priors by aggregating temporal variations through diffusion graph convolution (Li et al., 2018; Wu et al., 2019). Specifically, for each input time slice, we apply the diffusion convolution operation: $\mathbf{H}_{t,:,:} = \text{DiffGCN}(\mathbf{X}_{t,:,:}^{\text{emb}}, \boldsymbol{A})$, where $\mathbf{X}^{\text{emb}} \in \mathbb{R}^{T \times N \times d}$ is the series embedding and $\boldsymbol{A}$ is the adjacency matrix. To simulate potential changes in graph topology during the expansion stage, we further apply edge dropout to $\boldsymbol{A}$ during training, enhancing robustness to evolving structures. The final adaptive embedding is obtained by combining static refinement and dynamic aggregation:

$$(\mathbf{Z}_{\text{pmt}})_{t,:,:} = \boldsymbol{Z}_{\text{ref}} + \mathbf{H}_{t,:,:}. \tag{4}$$

This embedding $\mathbf{Z}_{\text{pmt}}$ not only incorporates static priors but also adapts to temporal variations, serving as the prompting within the forecasting model. This adjustment block, an MLP and a diffusion graph convolution, is trained jointly with the STF backbone in both base stage and expansion stage and is executed at test stage to produce the final prompts used by the predictor.

**Prompting initialization in expansion stage.** In the expansion stage, new nodes often lack sufficient history to compute reliable priors. For these nodes, we adopt a similarity-based initialization. Their priors can either be recomputed directly from the limited data available in the expansion stage, or constructed by weighted mixing of the priors from a few most similar remain nodes in the base stage. Similarity is measured using the cross-correlation matrix $\boldsymbol{P}$ introduced in Section 4.1.2, recomputed under the current stage. Formally, For a new node $i$, its similarity weight with remain node $j$ is calculated as $s_{i,j} = P_{i,j} / \sum_{j \in \mathcal{V}_{\text{remain}}} P_{i,j}, v_i \in \mathcal{V}_{\text{new}}, ; v_j \in \mathcal{V}_{\text{remain}}$. Let $\mathbb{P}_{\text{all}} = \{\text{prd}, \text{topo}, \text{tdi}\}$ denote the *candidate* prior types, and let $\mathbb{P}_i \subseteq \mathbb{P}_{\text{all}}$ be the subset actually constructed for new node $i$. For any prior type $\P \in \mathbb{P}_i$ with feature matrix $\boldsymbol{Z}_\P \in \mathbb{R}^{N \times d_\P}$, the prior of node $i$ is obtained by mixing the priors of its top similar remain nodes: $(\boldsymbol{z}_\P)_i = \sum_{j \in \mathcal{N}_{k(i)}} s_{i,j} (\boldsymbol{z}_\P)_j$, where $\mathcal{N}_{k(i)}$ denotes the top-$k$ most similar remain nodes to $i$.

Not all prior types in $\mathbb{P}_{\text{all}}$ require mixing. In practice, a simple similarity threshold determines whether a new node recomputes its priors or mixes them from similar remain nodes. In our experiments, periodic priors are mixed from old nodes due to insufficient cycle history, whereas topological and time-delayed interaction priors are recomputed from short-term observations because they reflect recent and rapidly varying spatial dependencies. Remain nodes simply reuse their base-stage priors. Most prior works do not address nodes removed in the expansion stage. In our framework, discarded nodes require no priors in the new period, and because model parameters are fully decoupled from node identity, no redundant parameters persist. This avoids parameter waste and enhances flexibility for evolving network structures.

## 4.3 INTEGRATION WITH SPATIAL-TEMPORAL FORECASTING MODELS

Based on the static priors and dynamic refinement introduced above, SNIP can be seamlessly integrated into existing STF architectures by injecting the prompting into the input features before

spatial-temporal feature extraction. To establish a baseline for expanding-node forecasting, we integrate the SNIP framework with a recent efficient spatio-temporal encoder (Zheng et al., 2025b), which provides a general mechanism for learning compact and expressive representations with complexity linear in the number of nodes. The resulting model, SNIPformer, incorporates our prior-guided prompting into the encoder's input embedding and spatial-temporal extraction process, followed by a lightweight regression head for prediction. Appendix A.3 details the complete model structure, the implementation of the prior, and the algorithms for base-stage pre-training and expansion-stage fine-tuning.

## 5 EXPERIMENTS

In this section, we evaluate and analysis the effectiveness, generality, and flexibility of our proposed SNIP framework under node expansion scenarios using four real-world datasets.

### 5.1 EXPERIMENT SETTING

**Datasets and Evaluation Setting.** We use the following spatial-temporal datasets across traffic and energy domain for evaluation: **EPeMS** (Ma et al., 2025b), **PEMS04** (Song et al., 2020), **SeaLoop** (Cui et al., 2019), and **NREL-AL** (Xu et al., 2025). For EPeMS, we follow the node expansion setup introduced in STEV (Ma et al., 2025b). For the other datasets, we simulate node expansion by randomly partitioning the node set into remain, deleted, and newadd groups. The detailed implementation procedure is provided in the Appendix B.1. Table 1 summarizes the stage and node partitions. We use a 12-step history to predict the next 12 steps, correspond to 1 hour ahead prediction. Beyond single-stage expansion, we also test SNIP in **multi-stage expanding-node scenarios** using the PEMS-Stream (Chen et al., 2021) and Air-Stream (Chen & Liang, 2025) datasets , where the node set evolves over several consecutive expansion periods. Moreover, SNIP is evaluated in **multi-horizon forecasting** (24/48/96-step) settings. Detailed results and analyses are provided in Appendix B.5 and B.6.

We compute prior features in the ***base stage*** using full historical data and train models with sliding-window samples. In the ***expansion stage***, priors are recomputed from short-term history and priors transferred from the base stage, followed by fine-tuning. Final evaluation is conducted in the ***test stage***. We report Mean Absolute Error (MAE) and Root Mean Squared Error (RMSE) in the main tables, while Mean Absolute Percentage

Table 1: Dataset statistics and characteristics

| Dataset | Stage Split $\tau_1$ / $\tau_2$ / test | Node Expansion $(\tau_1 \rightarrow \tau_2)$ |
|---|---|---|
| EPeMS | 63d / 3d + 2d / 22d | $296 \rightarrow 447$ |
| PEMS04 | 35d / 6d + 1d / 17d | $241 \rightarrow 290$ |
| SeaLoop | 18d / 6d + 1d / 3d | $255 \rightarrow 303$ |
| NREL-AL | 122d / 6d + 1d / 53.5d | $103 \rightarrow 130$ |

Error (MAPE) and Mean Relative Error (MRE) are provided in the Appendix B.4 with consistent conclusions.

**Baselines and Hyperparameter Settings.** We compare SNIPformer (introduced in Section 4.3) with four categories of existing solutions for expanding-node STF: 1) Models without node-specific prompting: **DLinear** (Zeng et al., 2023), **iTransformer** (Liu et al., 2024), **DUET** (Qiu et al., 2025). 2) STF models without node-specific modules: **DCRNN** (Li et al., 2018), **GWNET**[†] (Wu et al., 2019), **GMAN** (Zheng et al., 2020), **STID**[†] (Shao et al., 2022), **STAEformer**[†] (Liu et al., 2023), **TESTAM**[†] (Lee & Ko, 2024), **STOP** (Ma et al., 2025a), where † indicates removal of learnable node embeddings. 3) Continual learning methods: **STKEC** (Wang et al., 2023), **EAC** (Chen & Liang, 2025). 4) Fixed-node models after expansion: **STEV** (Ma et al., 2025b). For SNIPformer, we set the PCA feature dimension to 24 (each for daily and weekly periods) and the spectral embedding dimension to 8. The model dimension is 64 (32 for NREL-AL). Other implementation details are provided in the Appendix. Average results are reported after repeating the experiments no less than five times.

### 5.2 EFFECTIVENESS AND GENERALITY

**Expanding-node forecasting results.** Table 2 summarizes the results across all nodes, *Remain* nodes, and *New* nodes, where the best results are highlighted in **bold red** and the second-best results

Table 2: Comparison of the expanding-node forecasting results of different methods and SNIP-former.

| Model | Metric | EPeMS | | | PEMS04 | | | SeaLoop | | | NREL-AL | | |
|---|---|---|---|---|---|---|---|---|---|---|---|---|---|
| | | All | Remain | New | All | Remain | New | All | Remain | New | All | Remain | New |
| DLinear | MAE | 32.70 | 32.26 | 33.56 | 28.97 | 28.91 | 29.17 | 4.59 | 4.62 | 4.49 | 2.54 | 2.59 | 2.41 |
| | RMSE | 48.32 | 48.14 | 48.66 | 44.55 | 44.29 | 45.42 | 7.99 | 8.02 | 7.92 | 3.91 | 4.00 | 3.63 |
| iTransformer | MAE | 26.83 | 26.65 | 27.16 | 24.76 | 24.74 | 24.83 | 4.29 | 4.31 | 4.21 | 1.94 | 1.97 | 1.83 |
| | RMSE | 41.40 | 41.22 | 41.73 | 39.62 | 39.34 | 40.52 | 7.54 | 7.56 | 7.46 | 3.36 | 3.44 | 3.12 |
| DUET | MAE | 25.25 | 25.17 | 25.39 | 23.21 | 23.21 | 23.24 | 4.02 | 4.04 | 3.93 | 1.82 | 1.85 | 1.72 |
| | RMSE | 38.05 | 38.18 | 37.77 | 36.54 | 36.32 | 37.26 | 7.02 | 7.04 | 6.93 | 3.10 | 3.17 | 2.88 |
| GWNET[†] | MAE | 23.73 | 23.11 | 24.93 | 22.99 | 23.05 | 22.79 | 3.94 | 3.97 | 3.84 | 1.79 | 1.85 | 1.69 |
| | RMSE | 35.81 | 35.27 | 36.84 | 36.70 | 36.57 | 37.16 | 6.82 | 6.86 | 6.68 | 3.16 | 3.24 | 2.92 |
| STID[†] | MAE | 24.40 | 24.31 | 24.56 | 22.49 | 22.56 | 22.25 | 4.10 | 4.12 | 4.03 | 2.00 | 2.03 | 1.89 |
| | RMSE | 37.38 | 37.44 | 37.23 | 35.92 | 35.77 | 36.42 | 7.26 | 7.28 | 7.20 | 3.25 | 3.33 | 3.01 |
| STAEformer[†] | MAE | 24.86 | 24.66 | 25.27 | 22.95 | 23.03 | 22.67 | 4.15 | 4.17 | 4.09 | 1.91 | 1.95 | 1.81 |
| | RMSE | 38.34 | 38.26 | 38.50 | 36.75 | 36.63 | 37.14 | 7.38 | 7.39 | 7.37 | 3.29 | 3.37 | 3.05 |
| STOP | MAE | 24.45 | 24.47 | 24.41 | 22.54 | 22.56 | 22.46 | 4.12 | 4.13 | 4.08 | 2.01 | 2.05 | 1.89 |
| | RMSE | 37.24 | 37.41 | 36.89 | 35.74 | 35.52 | 36.45 | 7.32 | 7.32 | 7.35 | 3.25 | 3.32 | 3.02 |
| STKEC | MAE | 29.99 | 29.78 | 30.40 | 25.64 | 25.84 | 24.87 | 5.00 | 5.01 | 4.98 | 2.33 | 2.34 | 2.28 |
| | RMSE | 42.91 | 43.05 | 42.64 | 39.55 | 39.74 | 38.73 | 8.14 | 8.13 | 8.16 | 3.62 | 3.66 | 3.51 |
| EAC | MAE | 28.74 | 28.23 | 29.75 | 24.05 | 24.27 | 23.21 | 4.72 | 4.73 | 4.72 | 2.16 | 2.17 | 2.14 |
| | RMSE | 40.33 | 39.80 | 41.35 | 36.51 | 36.79 | 35.41 | 7.82 | 7.81 | 7.86 | 3.37 | 3.39 | 3.32 |
| STEV | MAE | _22.90_ | _22.35_ | _23.97_ | _20.55_ | _20.42_ | _21.01_ | _3.92_ | _3.95_ | _3.84_ | **1.57** | **1.58** | **1.53** |
| | RMSE | _34.51_ | _33.95_ | _35.60_ | _32.46_ | _32.13_ | _33.53_ | _6.62_ | _6.66_ | _6.51_ | **2.88** | **2.93** | **2.73** |
| SNIPformer (ours) | MAE | **22.05** | **21.39** | **23.35** | **19.20** | **19.22** | **19.10** | **3.46** | **3.47** | **3.42** | _1.62_ | _1.65_ | _1.55_ |
| | RMSE | **33.91** | **33.16** | **35.33** | **31.02** | **30.87** | **31.54** | **6.10** | **6.14** | **5.97** | _2.87_ | _2.92_ | _2.71_ |

in underlined blue. SNIP achieves the best performance on the three traffic datasets, with relative averaged improvements up to 7.61% / 5.61% in MAE and RMSE over the strongest baselines. On NREL-AL, SNIP ranks second on MAE, slightly below STEV. We attribute this gap to domain-specific characteristics, such as stronger trend strength (Qiu et al., 2024)(in Table 5) and more severe distribution shifts, which are more effectively captured by the contrastive learning strategy in STEV. Nevertheless, compared to node-agnostic models and continual learning approaches, SNIP consistently delivers superior accuracy, confirming the effectiveness of structured priors in encoding node heterogeneity under expansion scenarios. More results under the multi-stage expansion and multi-horizon settings are provided in Appendix B.5 and B.6.

**Generality across architectures.** To validate SNIP's model-agnostic design, we integrate it into four categories of backbones: MLP based (DLinear, STID), graph based (DCRNN, GWNET[†], GMAN), attention based (iTransformer, DUET, STAEformer[†]), and a hybrid architecture TESTAM[†], three experts for temporal modeling, static graph spatio-temporal modeling, and dynamic graph spatio-temporal modeling). Additionally, we employ an attention-based module from STGP (Hu et al., 2024) as a prompting baseline, referring to it as AttP in this experiment. Table 3 reports the MAE results of (i) the **original backbone**, (ii) **backbone + AttP**, and (iii) **backbone + SNIP** from three representative backbones. Full results are provided in Appendix B.4 (Table 10 and Table 11). Across all cases, AttP does not yield noticeable improvements, whereas SNIP consistently and significantly enhances forecasting performance under dynamic node changes. This confirms that prior-guided prompting provides a more effective way to capture node heterogeneity and adapt to evolving networks. More importantly, these results highlight SNIP's generality: as a model-agnostic framework, it can be seamlessly combined with diverse forecasting architectures, enabling them to remain effective in expanding-node scenarios while preserving strong accuracy. This suggests that prompting frameworks and spatio-temporal feature extractors can evolve in parallel as complementary directions.

Table 3: Forecasting MAE of different backbones with and without prompting modules.

| Model | EPeMS | | | NREL-AL | | |
|---|---|---|---|---|---|---|
| | All | Remain | New | All | Remain | New |
| iTransformer | 26.83 | 26.65 | 27.16 | 1.94 | 1.97 | 1.83 |
| + AttP | 26.81 | 26.64 | 27.14 | 1.95 | 1.99 | 1.84 |
| + SNIP | **24.67** | **24.14** | **25.71** | **1.84** | **1.88** | **1.74** |
| GWNET[†] | 23.73 | 23.11 | 24.93 | 1.79 | 1.83 | 1.69 |
| + AttP | 23.75 | 23.13 | 24.96 | 1.79 | 1.82 | 1.69 |
| + SNIP | **23.41** | **22.79** | **24.62** | **1.77** | **1.81** | **1.66** |
| STID[†] | 24.40 | 24.31 | 24.56 | 2.00 | 2.03 | 1.89 |
| + AttP | 24.35 | 24.26 | 24.53 | 2.01 | 2.05 | 1.90 |
| + SNIP | **21.84** | **21.13** | **23.23** | **1.86** | **1.89** | **1.77** |

## 5.3 ABLATION AND HYPER-PARAMETER STUDIES

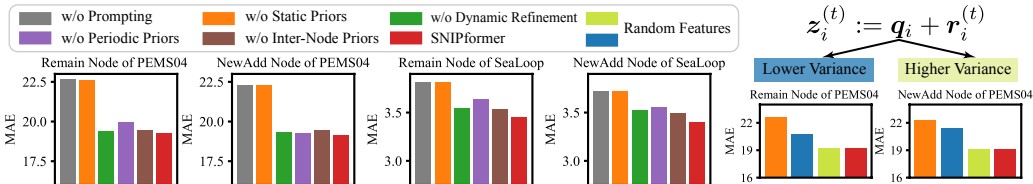

Figure 3: Ablation Results. Left: Comparison of contribution of different components. Right: Performance of using random features with high and low variance as static priors.

**Component-wise analysis.** We first assess the contribution of different components in SNIP by progressively removing them: (i) w/o Prompting, (ii) w/o Static Priors, (iii) w/o Dynamic Refinement, (iv) w/o Periodic Priors, and (v) w/o Inter-node Priors (removing both topology and time-delayed interaction priors). Figure 3 reports results on PEMS04 and SeaLoop, evaluated on *remain* nodes and *newadd* nodes. The results yield several key insights. Removing prompting causes a substantial accuracy drop; relying solely on dynamic refinement to learn full embeddings also performs poorly, suggesting that directly fitting optimal embeddings without helpful priors is highly challenging. In contrast, using only static priors without refinement underscores the necessity of modeling temporal variations. Finally, eliminating periodic or inter-node priors consistently degrades performance, validating that the constructed priors effectively encode node heterogeneity and structural dependencies.

**Empirical analysis of decomposition and heterogeneity.** We further validate Hypothesis 1 by replacing static priors with alternative designs: (a) random priors with high variance, (b) random priors with low variance, and (c) no static priors. Figure 4 shows that under the decomposition framework of Hypothesis 1, even randomly initialized features can achieve competitive results. Moreover, larger initialization variance improves performance, underscoring the importance of heterogeneity.

To intuitively demonstrate the heterogeneity introduced by SNIP prompting, we visualize results on the PEMS04 dataset under a fixed-node forecasting setup with STID in Figure 4. When the learnable embeddings in STID are either replaced by SNIP or augmented with SNIP, both heterogeneity score (Chen & Liang, 2025) and predictive performance improve. As shown in the t-SNE visualization under a unified embedding space, the combination of learnable embeddings and SNIP yields a wider spread and more distinct clusters, indicating that SNIP effectively enhances heterogeneity.

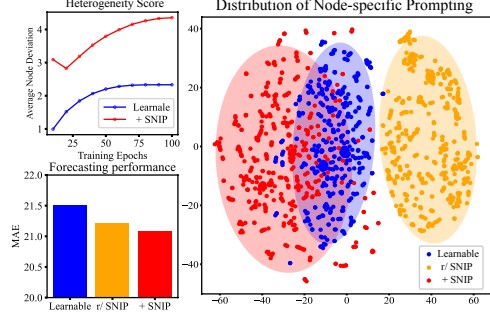

Figure 4: Contribution of SNIP to the STID model on PEMS04 dataset. Right: Distribution of node-specific prompting after dimensionality reduction via t-SNE.

**Hyper-parameter study.** We examine two groups of hyper-parameters that control prior construction: $k_{\text{pca}}$ and the number of periodicities $n$ for the periodic priors, and $k_{\text{topo}}, k_{\text{delay}}, k_{\text{corr}}$ together with the Hann window size of Welch method for the interaction priors. Figure 5 reports RMSE on PEMS04, and full results are given in Appendix B.3. In Figure 5(a), varying $k_{\text{pca}}$ shows that performance is stable across a wide range. We set $k_{\text{pca}} = 24$ to keep preprocessing cost low while retaining prediction accuracy. In addition, using both daily and weekly periods ($n = 2$) consistently outperforms using either alone, which matches the multi seasonal nature of traffic data. In Figure 7(b), increasing $k_{\text{topo}}, k_{\text{delay}}, k_{\text{corr}}$ beyond small values brings negligible gains but higher training and preprocessing cost. For the Hann window size, performance varies within a narrow band and stabilizes as the window grows. We set it to $T$ (12 in our task) in the main experiments as a balanced choice that matches the forecasting horizon and preserves time localization.

Moreover, we further evaluate different ratios of new and retired nodes to demonstrate robustness under diverse real deployment scenarios, with results reported in Appendix B.3. In addition, we

visualize node distributions and forecasting performance under three expansion protocols (following STEV and detailed in Appendix B.1), and present the corresponding results in Appendix B.7.

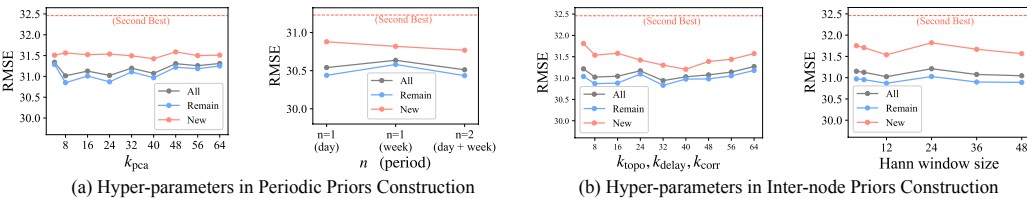

(a) Hyper-parameters in Periodic Priors Construction          (b) Hyper-parameters in Inter-node Priors Construction

Figure 5: Hyper-parameter study in PEMS04 dataset.

## 5.4 EFFICIENCY AND FLEXIBILITY

Computational efficiency is an important consideration for expanding-node forecasting. The additional cost of SNIP mainly comes from three preprocessing operations: multi-cycle PCA for periodic features, cross-correlation estimation between node pairs, and spectral embedding of the resulting matrices. Crucially, all of these steps are performed once in the base stage, and the priors are reused throughout training and expansion. As shown in Table 4, the one-off preprocessing overhead is minor compared with training time.

When comparing training and inference efficiency, SNIPformer shows clear advantages over the strongest baseline, STEV. While STEV incurs heavy retraining whenever nodes are expanded, SNIPformer requires only lightweight fine-tuning with precomputed priors. This results in substantial reductions in both training time and memory consumption, while maintaining competitive accuracy. In addition, applying SNIP to classical backbones such as STAEformer introduces only minimal extra cost, yet enables these models to operate effectively in expansion scenarios where their original designs fail. Overall, SNIP achieves high efficiency, flexibility, and scalability, offering a model-agnostic prompting framework that can be seamlessly incorporated into existing or future STF architectures.

Table 4: Training and inference efficiency comparison on EPeMS (batch size = 32).

| Metric | STEV | SNIPformer | ↕% | STAEformer | STAEformer[†] +SNIP | ↕% |
|---|---|---|---|---|---|---|
| **Pre-computation** Time Cost (min) | Augmentation 0.21 | Static Priors 2.61 | - | - | Static Priors 2.61 | - |
| **Training** Time (s/epoch) Footprint (MB) | $(\tau_1, \tau_2)$ 325.42 31430 | $(\tau_1 \rightarrow \tau_2)$ **28.26 → 1.42** **1466 → 2358** | ↓91.3% ↓92.5% | $(\tau_1)$ **132.03** **8130** | $(\tau_1)$ 134.77 8296 | ↑2.0% ↑2.0% |
| **Inference** Time(s) $(\tau_2)$ MAE | 20.46 22.90 | **1.05** **22.05** | ↓94.9% ↓3.7% | Invalid | 1.37 23.75 | ↑ Feasibility |

## 6 CONCLUSION

In this paper, we proposed SNIP, a model-agnostic prompting framework for expanding-node spatial-temporal forecasting. It constructs structured static priors from heterogeneity and correlation angles and performing learnable dynamic refinement. A similarity-weighted initialization further enables few-shot adaptation for new nodes. SNIP allows existing spatio-temporal forecasting models to be easily adapted to expanding-node scenarios. Experiments across multiple datasets and backbones show that SNIP achieves strong accuracy, generality, and efficiency. Ablations show that variance-preserving, correlation-aware priors and dynamic refinement are all indispensable. Future work will study the optimal composition of prompting, extend SNIP to cross-domain settings, and integrate it as a prompting layer in large spatial-temporal models.

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

## A  APPENDIX: METHODOLOGY DETAILS

### A.1  PERIODIC PRIORS CONSTRUCTION

Given a historical period $\tau$ with a length of $L$. Let $\boldsymbol{X} \in \mathbb{R}^{L \times N}$ denote the historical sequence of for $N$ nodes. We specify a set of cycle lengths $\{p_1, p_2, \ldots, p_n\}$. For node $i$ and a given cycle length $p_j$, we partition its sequence $\boldsymbol{X}_{:,i} \in \mathbb{R}^L$ into non-overlapping cycle segments. For example, when $p_j$

corresponds to one day, the sequence is divided into consecutive daily fragments, each treated as an individual segment. Formally, the set of segments is defined as:

$$\mathbb{S}_i^{(j)} = \text{Partition}(\boldsymbol{X}_{:,i},\, p_j) = \left\{ \boldsymbol{X}_{:,i}[(m-1)p_j + 1 : mp_j] \,\Big|\, m = 1, \ldots, M_j \right\}, \tag{5}$$

where $M_j = \lfloor L/p_j \rfloor$ is the number of complete cycles. Each element of $\mathbb{S}_i^{(j)}$ is a vector in $\mathbb{R}^{p_j}$. Before dimensionality reduction, each segment of node $i$ is normalized independently: $\tilde{\boldsymbol{x}} = (\boldsymbol{x} - \mu_i^{(j)})/\sigma_i^{(j)}$, $\boldsymbol{x} \in \mathbb{S}_i^{(j)}$, where $\mu_i^{(j)}$ and $\sigma_i^{(j)}$ are the mean and standard deviation of node $i$'s segments under cycle length $p_j$.

Each normalized segment $\tilde{\boldsymbol{X}}_m^{(j)}$ is treated as an $N \times p_j$ data matrix, which is the full "snapshot" of all nodes in cycle $j$. We then apply Principal Component Analysis (PCA) to reduce these segments to their low-rank components and obtain compact representations. Specifically, PCA yields a projection matrix $\boldsymbol{U}^{(j)} \in \mathbb{R}^{p_j \times k_{\text{pca}}}$ from the top $k_{\text{pca}}$ eigenvectors of the covariance matrix of $\tilde{\boldsymbol{X}}_m^{(j)}$. Then the segment-level low-dimensional representation is computed, and the $M_j$ representations are the averaged across segments:

$$\bar{\boldsymbol{Z}}^{(j)} = \frac{1}{M_j} \sum_{m=1}^{M_j} \tilde{\boldsymbol{X}}_m^{(j)} \boldsymbol{U}^{(j)} \in \mathbb{R}^{N \times k_{\text{pca}}}, j \in [1, ..., n]. \tag{6}$$

Finally, the representations from all $n$ cycle lengths are concatenated, yielding the periodic prior feature matrix:

$$\boldsymbol{Z}_{\text{prd}} = \text{Concat}\left( \bar{\boldsymbol{Z}}^{(1)}, \bar{\boldsymbol{Z}}^{(2)}, \ldots, \bar{\boldsymbol{Z}}^{(n)} \right) \in \mathbb{R}^{N \times (n \cdot k_{\text{pca}})}. \tag{7}$$

Figure 6 illustrates this construction.

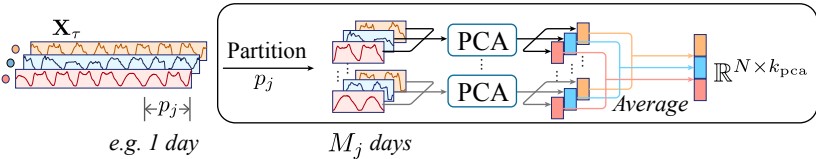

Figure 6: An illustration for the periodic prior construction under a cycle length $p_j$.

## A.2 TIME-DELAYED INTERACTION PRIORS CONSTRUCTION

As a increasing trend investigated by recent studies(Long et al., 2024; Zheng et al., 2025a), the correlation between two node sequences is often more pronounced when a temporal delay is considered rather than assuming synchronous dynamics. To capture this, we quantify their association through the cross power spectral density, which avoids the limitation of manually specifying a maximum delay as required in previous research. This formulation allows us to directly compute the delay step that maximizes their correlation, along with the corresponding strength. Intuitively, these two quantities characterize both the temporal span and the spatial extent of the interaction between nodes.

Formally, let $\boldsymbol{x}_i, \boldsymbol{x}_j \in \mathbb{R}^L$ denote the historical sequences of nodes $i$ and $j$. Each sequence is normalized in the same manner as in periodic features. Their cross-spectral density (CSD) is estimated using Welch's method with a Hann window (Welch, 1967) of length $T$:

$$Q_{ij}(\nu) = \frac{1}{K} \sum_{k=1}^{K} X_i^{(k)}(\nu) X_j^{(k)}(\nu)^*, \tag{8}$$

where $K = \lfloor L/T \rfloor$ is the number of windows, $X_i^{(k)}(\nu)$ is the Fourier transform of the $k$-th windowed segment of node $i$, $\nu$ is the frequency variable, and $^*$ denotes complex conjugation. The cross-correlation function is obtained by inverse FFT:

$$R_{ij}(\delta) = \mathcal{F}^{-1}\left(Q_{ij}(\nu)\right), \tag{9}$$

which is then shifted to align both positive and negative delays. We extract the most significant delay and its corresponding correlation strength as

$$\Delta_{ij} = \arg\max_\delta |R_{ij}(\delta)|, \qquad \boldsymbol{P}_{ij} = \max_\delta |R_{ij}(\delta)|. \tag{10}$$

where $\Delta$ is the delay matrix recording absolute dominant lags, and $\boldsymbol{P}$ is the correlation matrix recording absolute correlation strengths. Following the same procedure, we apply spectral embedding to the delay and correlation matrices, and then concatenate them into $\boldsymbol{Z}_{\text{tdi}}$:

$$\boldsymbol{Z}_{\text{tdi}} = \text{Concat}(\Phi(\Delta, k_{\text{delay}}), \Phi(\boldsymbol{P}, k_{\text{corr}})). \tag{11}$$

### A.3 ARCHITECTURE OF SNIPFORMER AND ALGORITHM OF PIPELINES

Figure 7 represents the entire architecture of SNIPformer. We use the data embedding module and spatial-temporal extractor proposed by ST-ReP (Zheng et al., 2025b) as the main architecture. Differently, we remove the learnable spatial embeddings in the original model and use our dynamic refinement module and pre-computed static priors to build a new node embedding for input series. Moreover, we use a linear head to transform the flattened spatial-temporal hidden features into prediction.

To facilitate reproducibility and implementation, we provide detailed algorithmic descriptions of the proposed SNIP framework, covering prior construction, base stage training, and expansion stage adaptation. These procedures are summarized in Algorithm 1, Algorithm 2, Algorithm 3, and Algorithum 4, respectively.

---

**Algorithm 1** OfflinePriorConstruction

---

**Input:** $\mathbf{X}_{\text{hist}} \in \mathbb{R}^{N \times L \times C}$, graph matrix $\boldsymbol{A} \in \mathbb{R}^{N \times N}$, config cfg
**Output:** $\boldsymbol{Z}_{\text{prd}}, \boldsymbol{Z}_{\text{topo}}, \boldsymbol{Z}_{\text{tdi}}$
  1: **Periodic prior via PCA**
  2: **for** $j = 1$ to $n$ **do**
  3:      $\boldsymbol{Z}_{\text{prd}}^{(j)} \leftarrow \text{SEGMENTPCA}(X_{\text{hist}}, p_j, k_{\text{pca}})$    $\triangleright \boldsymbol{Z}_{\text{prd}}^{(j)} \in \mathbb{R}^{N \times k_{\text{pca}}}$    $\triangleright p_j$ is the $j$-th cycle length
  4: **end for**
  5: $\boldsymbol{Z}_{\text{prd}} \leftarrow \text{concat}(\boldsymbol{Z}_{\text{prd}}^{(1)}, \dots, \boldsymbol{Z}_{\text{prd}}^{(n)})$
  6: $\boldsymbol{Z}_{\text{prd}} \in \mathbb{R}^{N \times (n \cdot k_{\text{pca}})}$

  7: **Topology prior via spectral embedding**
  8: $\boldsymbol{Z}_{\text{topo}} \leftarrow \text{SPECTRALEMBED}(\boldsymbol{A}, k_{\text{topo}})$

  9: **Time delayed interaction prior via spectral embedding**
 10: Initialize matrices $\Delta, \boldsymbol{P} \in \mathbb{R}^{N \times N}$ with zeros
 11: **for** each pair $(i, j)$ **do**
 12:      $Q_{ij} \leftarrow \text{WELCHCSD}(\mathbf{X}_{\text{hist}}[i, :], \mathbf{X}_{\text{hist}}[j, :], \text{Hann}, \text{cfg.nperseg}, \text{cfg.noverlap})$
 13:      $R_{ij} \leftarrow \text{ifft}(Q_{ij})$, real part, fftshift to centered lags
 14:      $\Delta[i, j] \leftarrow \arg\max_\delta |R_{ij}(\delta)|$
 15:      $\boldsymbol{P}[i, j] \leftarrow \max_\delta |R_{ij}(\delta)|$
 16: **end for**
 17: $\boldsymbol{Z}_{\text{tdi}} \leftarrow \text{concat}(\text{SPECTRALEMBED}(\Delta, k_{\text{delay}}), \text{SPECTRALEMBED}(\boldsymbol{P}, k_{\text{corr}}))$

 18: **return** $\boldsymbol{Z}_{\text{prd}}, \boldsymbol{Z}_{\text{topo}}, \boldsymbol{Z}_{\text{tdi}}$

---

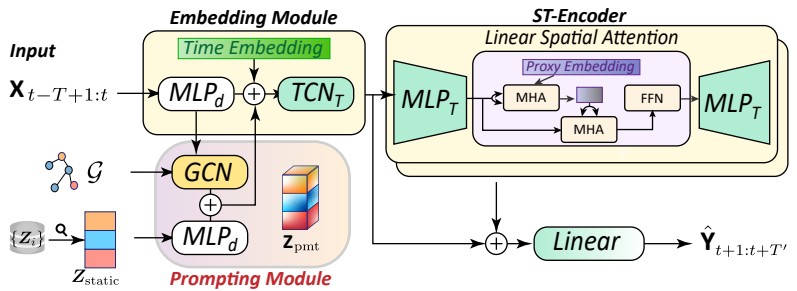

Figure 7: The architecture of SNIPformer.

---

**Algorithm 2** BaseStagePretrain

---

**Input:** base windows $\{(X_{t-L+1:t}, Y_t)\}$, priors $\boldsymbol{Z}_{\text{prd}}, \boldsymbol{Z}_{\text{topo}}, \boldsymbol{Z}_{\text{tdi}}$, backbone $f_\theta$, adjuster $g_\phi$
**Output:** trained parameters $\theta^\star, \phi^\star$

1: Initialize $\theta, \phi$
2: $\boldsymbol{Z}_{\text{static}} = Concat(\boldsymbol{Z}_{\text{prd}}, \boldsymbol{Z}_{\text{topo}}, \boldsymbol{Z}_{\text{tdi}})$
3: **for** each mini batch $\mathcal{B}$ **do**
4: $\quad \boldsymbol{Z}_{\text{pmt}} \leftarrow g_\phi(\boldsymbol{Z}_{\text{static}}, X_{t-L+1:t})$ $\qquad\qquad\qquad$ ▷ $g_\phi$ includes MLP and GCN layers
5: $\quad \hat{Y}_{t+1:t+T'} \leftarrow f_\theta(X_{t-T+1:t}, \boldsymbol{Z}_{\text{pmt}})$
6: $\quad \mathcal{L}_{\text{pred}} \leftarrow \ell(\hat{Y}_{t+1:t+T'}, Y_{t+1:t+T'})$
7: $\quad$ Update $\theta, \phi$ by minimizing $\mathcal{L}_{\text{pred}}$
8: **end for**
9: **return** $\theta^\star \leftarrow \theta, \ \phi^\star \leftarrow \phi$

---

**Algorithm 3** ExpansionStageAdapt and Inference

---

**Input:** expansion data, graph matrix $\boldsymbol{A}_{\tau_2}$, remain set $\mathcal{V}_{\text{rem}}$, new set $\mathcal{V}_{\text{new}}$,
$\qquad$ cached priors $\boldsymbol{Z}_{\text{prd}}, \boldsymbol{Z}_{\text{topo}}, \boldsymbol{Z}_{\text{tdi}}$, trained params $\theta^\star, \phi^\star$,
$\qquad$ mixing flags $(b_{\text{prd}}, b_{\text{topo}}, b_{\text{delay}}) \in \{0,1\}^3$, hyperparams $k_{\text{pca}}, k_{\text{topo}}, k_{\text{delay}}$,
$\qquad$ periodic set $\{p_j\}$, top $k$, similarity function $s(\cdot, \cdot)$
**Output:** adapted params $\theta^\dagger, \phi^\dagger$, updated priors $\boldsymbol{Z}_{\text{prd}}^\dagger, \boldsymbol{Z}_{\text{topo}}^\dagger, \boldsymbol{Z}_{\text{tdi}}^\dagger$

1: **Update static priors at expansion stage**
2: $\boldsymbol{Z}_{\text{prd}}^\dagger, \boldsymbol{Z}_{\text{topo}}^\dagger, \boldsymbol{Z}_{\text{tdi}}^\dagger \leftarrow \text{EXPANSIONPRIORSUPDATE}(\text{expansion data}, \boldsymbol{A}_{\tau_2}, \mathcal{V}_{\text{rem}}, \mathcal{V}_{\text{new}},$
$\qquad\qquad\qquad\qquad \boldsymbol{Z}_{\text{prd}}, \boldsymbol{Z}_{\text{topo}}, \boldsymbol{Z}_{\text{tdi}}, b_{\text{prd}}, b_{\text{topo}}, b_{\text{delay}},$
$\qquad\qquad\qquad\qquad \{p_j\}, k_{\text{pca}}, k_{\text{topo}}, k_{\text{delay}}, k, s)$ $\qquad$ ▷ in Algorithm 4
3: **Concatenate priors and initialize parameters**
4: $\boldsymbol{Z}_{\text{static}}^\dagger \leftarrow \text{concat}[\boldsymbol{Z}_{\text{prd}}^\dagger, \boldsymbol{Z}_{\text{topo}}^\dagger, \boldsymbol{Z}_{\text{tdi}}^\dagger]$
5: $\theta^\dagger \leftarrow \theta^\star, \quad \phi^\dagger \leftarrow \phi^\star$
6: **Expansion stage finetuing**
7: **for** each mini batch $\mathcal{B}$ from expansion windows **do**
8: $\quad \boldsymbol{Z}_{\text{pmt}}^\dagger \leftarrow g_{\phi^\dagger}(\boldsymbol{Z}_{\text{static}}^\dagger, \boldsymbol{X}_{t-L+1:t})$
9: $\quad \hat{\boldsymbol{Y}}_t \leftarrow f_{\theta^\dagger}(\boldsymbol{X}_{t-L+1:t}, \boldsymbol{Z}_{\text{pmt}}^\dagger)$
10: $\quad$ Update $\phi^\dagger$ and $\theta^\dagger$ by loss $\mathcal{L}_{\text{pred}}$
11: **end for**
12: **Inference**
13: **for** each test window **do**
14: $\quad \boldsymbol{Z}_{\text{pmt}}^\dagger \leftarrow g_{\phi^\dagger}(\boldsymbol{Z}_{\text{static}}^\dagger, \boldsymbol{X}_{t-L+1:t})$
15: $\quad$ Output $f_{\theta^\dagger}(\boldsymbol{X}_{t-L+1:t}, \boldsymbol{Z}_{\text{pmt}}^\dagger)$
16: **end for**
17: **return** $\theta^\dagger, \phi^\dagger, \boldsymbol{Z}_{\text{prd}}^\dagger, \boldsymbol{Z}_{\text{topo}}^\dagger, \boldsymbol{Z}_{\text{tdi}}^\dagger$

---

---

**Algorithm 4** ExpansionPriorsUpdate

---

**Input:** expansion data, graph matrix $\boldsymbol{A}_{\tau_2}$, remain set $\mathcal{V}_{\text{rem}}$, new set $\mathcal{V}_{\text{new}}$, cached priors $\boldsymbol{Z}_{\text{prd}}, \boldsymbol{Z}_{\text{topo}}, \boldsymbol{Z}_{\text{tdi}}$,
    mixing flags $(b_{\text{prd}}, b_{\text{topo}}, b_{\text{delay}}) \in \{0,1\}^3$, periodic set $\{p_j\}$, dims $k_{\text{pca}}, k_{\text{topo}}, k_{\text{delay}}$,
    top $k$, similarity function $s(\cdot, \cdot)$
**Output:** updated priors $\boldsymbol{Z}_{\text{prd}}^{\dagger}, \boldsymbol{Z}_{\text{topo}}^{\dagger}, \boldsymbol{Z}_{\text{tdi}}^{\dagger}$

1: **Partition expansion data**
2: $\mathbf{X}_{\text{rem}} \leftarrow \text{gather}(\text{expansion data}, \mathcal{V}_{\text{rem}})$, $\mathbf{X}_{\text{new}} \leftarrow \text{gather}(\text{expansion data}, \mathcal{V}_{\text{new}})$

3: **Similarity matrix and top-$k$ weights**
4: $\boldsymbol{S} \in \mathbb{R}^{|\mathcal{V}_{\text{new}}| \times |\mathcal{V}_{\text{rem}}|} \leftarrow \text{SIMILARITYMATRIX}(\mathbf{X}_{\text{new}}, \mathbf{X}_{\text{rem}}, s)$
5: $\boldsymbol{M} \leftarrow \text{TOPKMASK}(\boldsymbol{S}, k)$              ▷ row wise top $k$ indicator
6: $\alpha \leftarrow \text{ROWNORMALIZE}(\boldsymbol{S} \odot \boldsymbol{M})$      ▷ $\sum_j \alpha_{ij} = 1$ where $\boldsymbol{M}_{ij} = 1$

7: **Periodic prior update**
8: **if** $b_{\text{prd}} = 0$ **then**
9:      $\boldsymbol{Z}_{\text{prd}}^{\dagger} \leftarrow \text{SEGMENTPCA}(\text{expansion data}, \{p_j\}, k_{\text{pca}})$
10: **else**
11:      $\boldsymbol{Z}_{\text{prd}}^{\dagger}[\mathcal{V}_{\text{rem}}, :] \leftarrow \boldsymbol{Z}_{\text{prd}}[\mathcal{V}_{\text{rem}}, :]$, $\boldsymbol{Z}_{\text{prd}}^{\dagger}[\mathcal{V}_{\text{new}}, :] \leftarrow \alpha \cdot \boldsymbol{Z}_{\text{prd}}[\mathcal{V}_{\text{rem}}, :]$
12: **end if**

13: **Topology prior update**
14: **if** $b_{\text{topo}} = 0$ **then**
15:      $\boldsymbol{Z}_{\text{topo}}^{\dagger} \leftarrow \text{SPECTRALEMBED}(\boldsymbol{A}_{\tau_2}, k_{\text{topo}})$
16: **else**
17:      $\boldsymbol{Z}_{\text{topo}}^{\dagger}[\mathcal{V}_{\text{rem}}, :] \leftarrow \boldsymbol{Z}_{\text{topo}}[\mathcal{V}_{\text{rem}}, :]$,
18:      $\boldsymbol{Z}_{\text{topo}}^{\dagger}[\mathcal{V}_{\text{new}}, :] \leftarrow \alpha \cdot \boldsymbol{Z}_{\text{topo}}[\mathcal{V}_{\text{rem}}, :]$
19: **end if**

20: **Time delayed interaction prior update**
21: **if** $b_{\text{delay}} = 0$ **then**
22:      $\tilde{\Delta}, \tilde{\boldsymbol{P}} \leftarrow \text{BUILDDELAYCORRMATRIX}(\text{expansion data})$      ▷ Welch CSD method
23:      $\boldsymbol{Z}_{\text{tdi}}^{\dagger} \leftarrow \text{concat}(\text{SPECTRALEMBED}(\tilde{\Delta}, k_{\text{delay}}), \text{SPECTRALEMBED}(\tilde{\boldsymbol{P}}, k_{\text{delay}}))$
24: **else**
25:      $\boldsymbol{Z}_{\text{tdi}}^{\dagger}[\mathcal{V}_{\text{rem}}, :] \leftarrow \boldsymbol{Z}_{\text{tdi}}[\mathcal{V}_{\text{rem}}, :]$, $\boldsymbol{Z}_{\text{tdi}}^{\dagger}[\mathcal{V}_{\text{new}}, :] \leftarrow \alpha \cdot \boldsymbol{Z}_{\text{tdi}}[\mathcal{V}_{\text{rem}}, :]$
26: **end if**
27: **return** $\boldsymbol{Z}_{\text{prd}}^{\dagger}, \boldsymbol{Z}_{\text{topo}}^{\dagger}, \boldsymbol{Z}_{\text{tdi}}^{\dagger}$

---

# B APPENDIX: EXPERIMENT DETAILS

## B.1 DATASETS AND EVALUATION SETTING

We use the following spatial-temporal datasets across traffic and energy domain for evaluation:

- **EPeMS**(Ma et al., 2025b): an expansion-node dataset constructed in STEV (Ma et al., 2025b) from District 7 of California, which assumes no deleted nodes.
- **PEMS04** (Song et al., 2020): traffic flow data collected from the Caltrans Performance Measurement System in California.
- **SeaLoop** (Cui et al., 2019): Seattle traffic loop detector data, recording speed measurements.
- **NREL-AL** (Xu et al., 2025): renewable energy data, recording solar power generation from photovoltaic plants in Alabama in 2016.

The number of feature values for all dataset records is 1, i.e., $C = 1$.

**Stage and Node Division.** Each dataset is divided into three stages: a *base stage*, an *expansion stage*, and a *test stage*. Within the expansion stage, we further split the last portion (e.g., 1 day) as

the validation set, while the earlier portion (e.g., 6 days) is used for expansion-stage training. For EPeMS, we strictly follow the experimental setup in Ma et al. (2025b) for consistency. For the other datasets, 80% of nodes are randomly selected as observed nodes in the base stage, providing sufficient history ($L_1$). The remaining 20% are treated as **newadd** nodes, appearing only in the expansion stage with short history ($L_2 \ll L_1$). Additionally, 5% of base nodes are randomly designated as deleted, while the rest remain as **remain** nodes. Table 5 summarizes detail statistics of datasets.

Although these real-world spatial-temporal datasets provide sensor locations or physical adjacency, none of them include authoritative deployment or decommission timestamps, and, to the best of our knowledge, there is currently no publicly available benchmark that reflects truly incremental sensor deployments. This limitation has also been emphasized in STEV, which constructs expanding-node settings through spatial or internal partitioning due to the absence of real EVTS logs.

Following this practice, for EPeMS we adopt the same **area-expansion** protocol in STEV (Ma et al., 2025b), where new nodes correspond to sensors hypothetically deployed in newly covered regions. For the remaining datasets, despite available spatial coordinates or topological links, installation and retirement records are not provided. Therefore, we simulate the **internal expansions** in STEV, assigning a subset of nodes as remain, deleted, and newadd, and restricting the observation horizon of new nodes to emulate short post-deployment histories. This protocol follows the commonly adopted assumption that adding or removing sensors does not change the underlying physical process being monitored, making such synthetic expansions a reasonable proxy for evolving networks.

Looking forward, we believe that constructing benchmarks with true deployment logs or continuous sensor rollouts represents an important direction for the community. Such datasets would provide more realistic evaluation settings and further facilitate research on expanding-node spatial-temporal forecasting.

Table 5: Dataset statistics and characteristics

| Dataset | Sample Rate | Stage Split | Node Expansion ($\tau_1 \rightarrow \tau_2$) | Trend Strength |
|---------|-------------|-------------|----------------------------------------------|----------------|
| EPeMS | 5min | 63d / 3d + 2d / 22d | $296 \rightarrow 447$ (296 -0 + 151) | 0.12 |
| PEMS04 | 5min | 35d / 6d + 1d / 17d | $241 \rightarrow 290$ (241 -17 + 66) | 0.08 |
| SeaLoop | 5min | 18d / 6d + 1d / 3d | $255 \rightarrow 303$ (255 -20 + 68) | 0.11 |
| NREL-AL | 5min | 122d / 6d + 1d / 53.5d | $103 \rightarrow 130$ (103 -7 + 34) | 0.71 |

We additionally adopt two streaming benchmarks, Air-Stream and PEMS-Stream, which are commonly used in continual spatio temporal forecasting, to assess the performance of SNIP under multiple expansion phases:

- Air-Stream (Chen & Liang, 2025): an air quality index dataset introduced in the EAC (Chen & Liang, 2025) work, constructed from observations recorded at environmental monitoring stations across China, and it models a growing sensor network without node deletions.
- PEMS-Stream (Chen et al., 2021): a traffic flow dataset from the Traffic-Stream (Chen et al., 2021) work, built from District 3 in California, which assumes no deleted nodes.

Both datasets contain multiple node expansion stages and Air-Stream involves more than one thousand sensors, which allows evaluation at larger spatial scales. Detailed statistics of these streaming datasets are summarized in Table 6.

Table 6: Details of stream datasets with multi-stage node expansion

| Dataset | Sample Rate | Stage | Node Expansion | Average Growth Rate | Frames |
|---------|-------------|-------|----------------|---------------------|--------|
| Air-Stream | 1 hour | 4 | $1087 \rightarrow 1154$ $\rightarrow 1193 \rightarrow 1202$ | 3.43% | $8578 \rightarrow 8619$ $\rightarrow 8378 \rightarrow 8490$ |
| PEMS-Stream | 5 min | 7 | $655 \rightarrow 715 \rightarrow 786$ $\rightarrow 822 \rightarrow 834$ $\rightarrow 850 \rightarrow 871$ | 4.92% | $8928 \rightarrow 8928 \rightarrow 8928$ $\rightarrow 8928 \rightarrow 8928$ $\rightarrow 8928 \rightarrow 8928$ |

## B.2    BASELINE AND HYPER-PARAMETERS

We compare SNIPformer with four categories of existing solutions for expanding-node STF:

1. Models without node-specific prompting: **DLinear** (Zeng et al., 2023), **iTransformer** (Liu et al., 2024), **DUET** (Qiu et al., 2025).

2. STF models without node-specific modules: **DCRNN** (Li et al., 2018), **GWNET**[†] (Wu et al., 2019), **GMAN** (Zheng et al., 2020), **STID**[†] (Shao et al., 2022), **STAEformer**[†] (Liu et al., 2023), **TESTAM**[†] (Lee & Ko, 2024), **STOP** (Ma et al., 2025a), where † indicates removal of learnable node embeddings.

3. Continual learning methods: **STKEC** (Wang et al., 2023), **EAC** (Chen & Liang, 2025).

4. Fixed-node models after expansion: **STEV** (Ma et al., 2025b).

For SNIPformer, we set the PCA feature dimension to 24 (each for daily and weekly periods) and the spectral embedding dimension to 8. This leads to $k_{\mathrm{pca}} = 24$, $n = 2$, $k_{\mathrm{topo}} = k_{\mathrm{delay}} = k_{\mathrm{corr}} = 8$. Collectively, the dimension of $\boldsymbol{Z}_{\mathrm{static}}$ is 72. The hyper-parameters study is in Appendix B.3. During the expansion stage, the periodic priors of new nodes are constructed by mixing those of their three most similar remain nodes. Other priors are recomputed directly from the available expansion-stage data, except for the NREL-AL dataset, where the time-delayed interaction priors of new nodes are also obtained via mixing from old nodes. These means $\mathbb{P}_i = \{\mathrm{prd}\}$ for EPeMS, PEMS04 and SeaLoop datasets, while $\mathbb{P}_i = \{\mathrm{prd}, \mathrm{tdi}\}$ for NREL-AL dataset. These design choices are made in accordance with the degree of temporal distribution shift observed in each dataset.

We use a 12-step history to predict the next 12 steps, correspond to 1 hour ahead prediction, which denotes $T = T' = 12$. The model dimension is 64 (32 for NREL-AL). Average results are reported after repeating the experiments no less than five times. **Code and data source are provided in the Supplementary Material**. Our experiments is under the PyTorch framework on a Linux server with one Intel(R) Xeon(R) Gold 5220 CPU and one 32GB NVIDIA Tesla V100-SXM2 GPU card.

For methods where embeddings increase with expansion (i.e., continual learning approaches) or rely on fixed node-specific learnable parameters (e.g., STEV), the case of deleted nodes is not explicitly considered. In our implementation on datasets with node removals, we carefully align the learnable embeddings across stages. This means that the parameters corresponding to deleted nodes are also discarded during the expansion stage, ensuring fair and consistent evaluation.

## B.3    FULL RESULTS ON HYPER-PARAMETERS STUDY

We report a hyperparameter sensitivity study. The analysis is organized into two groups of design choices that correspond to the construction of the periodic prior and the node interaction priors. For each group, a grid search is carried out on PEMS04 and the MAE and RMSE curves are visualized in the sensitivity plots, while all other settings are fixed to their default values.

First, the hyperparameters for the periodic prior are examined, namely the number of periodic PCA components $k_{\mathrm{pca}}$ and the number of explicit periods $n$. Results are in Figure 8(a). The parameter $k_{\mathrm{pca}}$ controls the dimensionality of the periodic PCA subspace that summarizes day level and week level seasonal patterns. The sensitivity plots show that both MAE and RMSE remain in a narrow band when $k_{\mathrm{pca}}$ varies over a wide range. There is a relatively flat region around $k_{\mathrm{pca}} \in [16, 40]$, where accuracy is slightly better and the variance across runs is small, which indicates that the model does not critically depend on a specific choice as long as enough variance is retained by the components. To balance accuracy with preprocessing cost, $k_{\mathrm{pca}} = 24$ is used in the main experiments. For the number of periods $n$, three settings are compared, one daily cycle, one weekly cycle, and the combination of daily and weekly cycles. The plots confirm that using a single period loses useful structure, while combining day and week consistently yields the lowest MAE and RMSE, in line with the multi seasonal behavior observed in datasets.

Second, the hyperparameters for the node interaction priors are studied, namely the embedding dimensions $k_{\mathrm{topo}}, k_{\mathrm{delay}}, k_{\mathrm{corr}}$ and the Hann window size in the Welch estimator used for time delayed interactions. Results are in Figure 8(b). The three interaction dimensions control the latent spaces for the topology prior, the time delay prior, and the correlation prior. To keep these subspaces balanced when concatenated, they are tied to a common value and swept jointly. The sensitivity

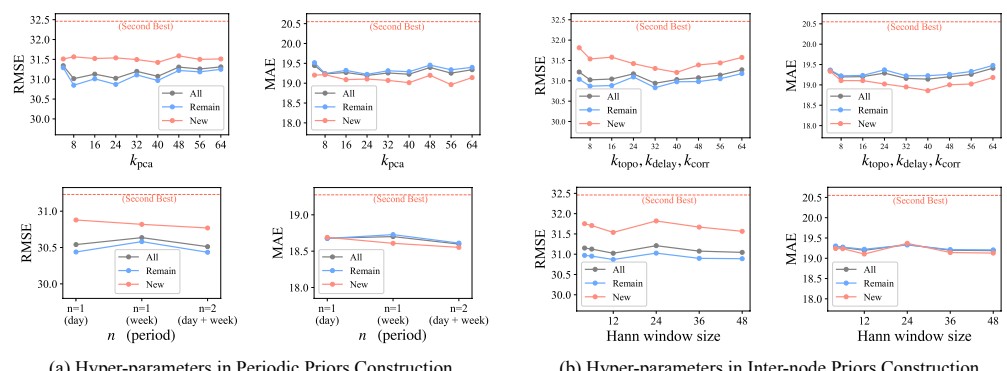

(a) Hyper-parameters in Periodic Priors Construction

(b) Hyper-parameters in Inter-node Priors Construction

Figure 8: Full results of the hyper-parameter study on PEMS04 dataset.

plots show only mild fluctuations of MAE and RMSE over the full range of embedding sizes. Very small dimensions slightly hurt performance, while very large dimensions provide no visible gain but increase both prior construction cost and refinement cost. A moderate setting with $k_{topo} = k_{delay} = k_{corr} = 8$ achieves a good trade off and is adopted as the default. For the Hann window size, the curves exhibit the expected bias variance behavior. Very short windows suffer from higher variance and weaker frequency resolution, which slightly degrades accuracy. As the window size increases, performance quickly enters a stable band and then saturates. We set the Hann window size to $T = 12$, which matches the forecasting horizon, providing a balanced compromise between variance reduction and time localization.

Finally, robustness with respect to network evolution is evaluated by varying the proportion of newly added nodes and the retirement rate of existing nodes. Figure 9 reports results for different new node ratios and retirement ratios. We observe that MAE and RMSE only change within a narrow range across all tested configurations. This indicates that the periodic prior and the interaction priors act as effective regularizers when the composition of the node set shifts, and that the model can gracefully handle a wide range of expansion profiles.

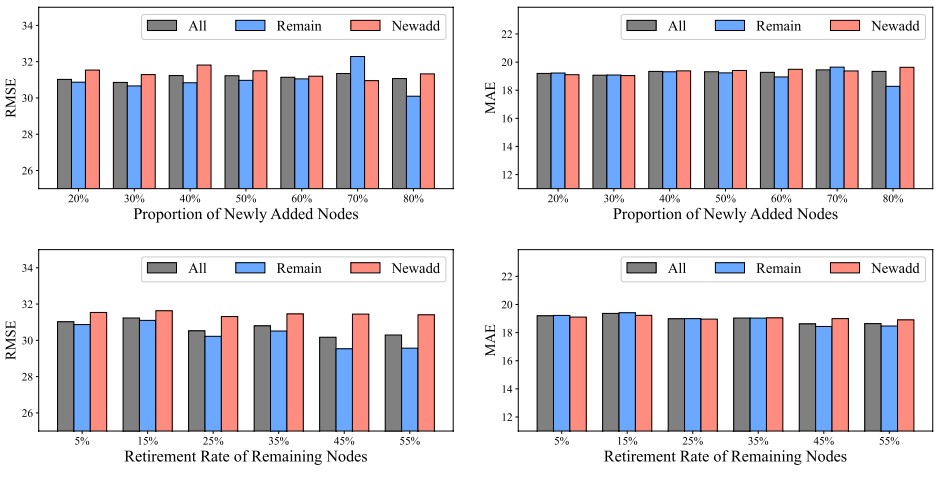

Figure 9: Full results under different node-division-ratio settings on PEMS04 dataset.

## B.4 FULL RESULTS ON EFFECTIVENESS AND GENERALITY

Table 9 reports the full forecasting results of the effectiveness study. SNIPformer achieves consistently the best accuracy on EPeMS, PEMS04, SeaLoop datasets and has a second-best performance on NREL-AL dataset. Compared with the strongest baseline STEV, the gains of SNIP arise from complementary representation and adaptation. STEV learns node specific parameters from short prediction windows, so each sample covers only a limited temporal context and struggles to encode persistent node heterogeneity. SNIP instead builds node identities as priors from the full historical record using multi period PCA, yielding low dimensional, stable, and discriminative prompts that preserve inter node differences while remaining independent of training window length. On top of these priors, SNIP employs a lightweight refinement module that learns corrections conditioned on the current short window and the diffusion context, which shifts the objective from fitting an all purpose embedding to adjusting residuals under changing conditions. When nodes are long term distinct but become temporarily similar due to events such as congestion, the priors anchor identity and the refinement adapts to the current regime. This combination translates into stronger generalization for new nodes with scarce history and more stable accuracy for remain nodes across horizons, aligning with the observed improvements over STEV.

Table 10 and Table 11 report the full forecasting results of the generality study. Taken together, these results indicate that, across MLP, graph, attention, and hybrid backbones, inserting SNIP consistently improves performance on All, Remain, and New nodes, which supports the interpretation of SNIP as a model agnostic prompting layer rather than a backbone specific trick. In comparison with the attention based prompting strategy (AttP), SNIP leverages explicitly constructed priors and flexible learned fine tuning to produce more informative node discriminative prompts and superior predictive accuracy, while remaining a plug and play component for architectures whose key layers are not hard tied to the cardinality of the node set.

## B.5 EVALUATION ON MULTIPLE EXPANSION STAGES

In addition to the fixed four stage settings, we further evaluate scalability and adaptability on two standard streaming benchmarks from EAC (Chen & Liang, 2025) with larger and more dynamic node sets. PEMS-Stream contains traffic data from 2011 to 2017, where the number of sensors increases from 655 to 871 over multiple expansion phases, and Air-Stream contains air quality measurements from 2016 to 2019 with more than one thousand sensors. Both benchmarks are designed for continual phase wise evaluation. Following the EAC protocol, we conduct 12 step ahead forecasting and report MAE and RMSE for SNIPformer, the lightweight backbone with SNIP (STID†+SNIP), and strong continual baselines EAC and STKEC, as summarized in Table 7 and Table 8. Across all phases on both datasets, SNIPformer and STID†+SNIP consistently outperform the continual baselines, which provides additional evidence that the proposed priors and prompting mechanism scale to longer and larger streams and remain effective under multiple successive expansion phases.

Table 7: Forecasting performance on PEMS-Stream dataset.

| Model | Metric | 2011 | 2012 | 2013 | 2014 | 2015 | 2016 | 2017 |
|---|---|---|---|---|---|---|---|---|
| STKEC | MAE | 15.80 | 15.77 | 15.86 | 16.77 | 16.27 | 15.64 | 17.16 |
| | RMSE | 24.63 | 25.00 | 25.96 | 27.60 | 26.85 | 27.91 | 28.17 |
| EAC | MAE | 14.51 | 14.23 | 14.37 | 15.20 | 14.87 | 14.28 | 15.91 |
| | RMSE | 22.22 | 22.14 | 23.13 | 24.31 | 24.26 | 25.65 | 25.87 |
| SNIPformer | MAE | **11.79** | **11.20** | **11.20** | 11.94 | 11.58 | 11.07 | 12.76 |
| | RMSE | **18.02** | **17.73** | 18.47 | **19.24** | **19.37** | 21.51 | 21.55 |
| STID†+SNIP | MAE | 12.30 | 11.29 | 11.23 | **11.59** | **11.51** | **10.86** | **12.62** |
| | RMSE | 19.20 | 18.34 | **18.18** | 19.40 | 19.58 | **21.23** | **21.18** |

Table 8: Forecasting performance on Air-Stream dataset.

| Model | Metric | 2016 | 2017 | 2018 | 2019 |
|-------|--------|------|------|------|------|
| STKEC | MAE | 31.04 | 27.04 | 20.16 | 21.46 |
|       | RMSE | 49.91 | 41.75 | 34.08 | 33.28 |
| EAC | MAE | 31.39 | 25.75 | 20.71 | 21.25 |
|     | RMSE | 49.99 | 39.34 | 34.20 | 32.94 |
| SNIPformer | MAE | 25.55 | 23.38 | 20.56 | 19.19 |
|            | RMSE | 41.31 | 36.80 | 35.35 | 30.50 |
| STID$^\dagger$+SNIP | MAE | **24.55** | **21.92** | **19.00** | **18.84** |
|                     | RMSE | **40.19** | **35.17** | **32.47** | **30.26** |

## B.6 EVALUATION ON MULTIPLE HORIZON SETTINGS

To further assess robustness under long range forecasting, we conduct multi horizon experiments on EPeMS using a lightweight backbone, STID$^\dagger$, with and without SNIP. Specifically, we consider four multi horizon configurations, where the most recent 12, 24, 48, and 96 time steps are used to predict the subsequent 12, 24, 48, and 96 time steps, respectively, and report MAE and RMSE for All, Remain, and New nodes, as summarized in Figure 10. Across all horizons and node groups, inserting SNIP consistently improves performance, with relative gains that generally increase from short to medium horizons and exhibit only a modest attenuation at 96 step forecasting. These results indicate that SNIP remains effective under multi step forecasting and does not degrade at longer horizons. This behavior is consistent with the design of the priors, where the periodic prior captures stable long cycle statistics to provide node specific identity prompts, while the dynamic refinement adjusts these prompts using short window time delayed interaction features, which helps mitigate potential drift of purely structural priors as the prediction horizon grows.

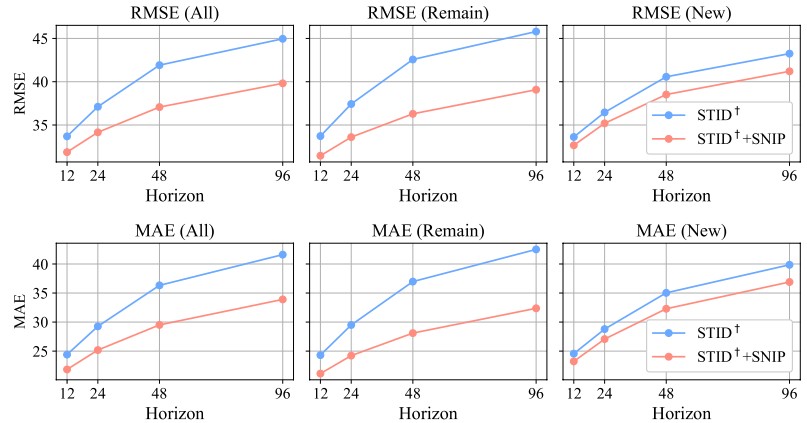

Figure 10: Evaluation on multi-horizon prediction.

## B.7 VISUALIZATION STUDY

Finally, we conduct a qualitative case study that links the geographic layout of node expansion to predictive performance. Figure 11 visualizes three representative scenarios: (a) area expansion on the EPeMS dataset, where new sensors appear in previously uncovered regions of the network, (b) spatial expansion on the EPeMS dataset, where the network is extended along existing corridors, and (c) internal expansion on the SeaLoop dataset, where additional sensors are inserted inside an already monitored region. In each map, existing sensors and newly added sensors are marked separately, which illustrates the different spatial patterns underlying the area, spatial, and internal expansion regimes.

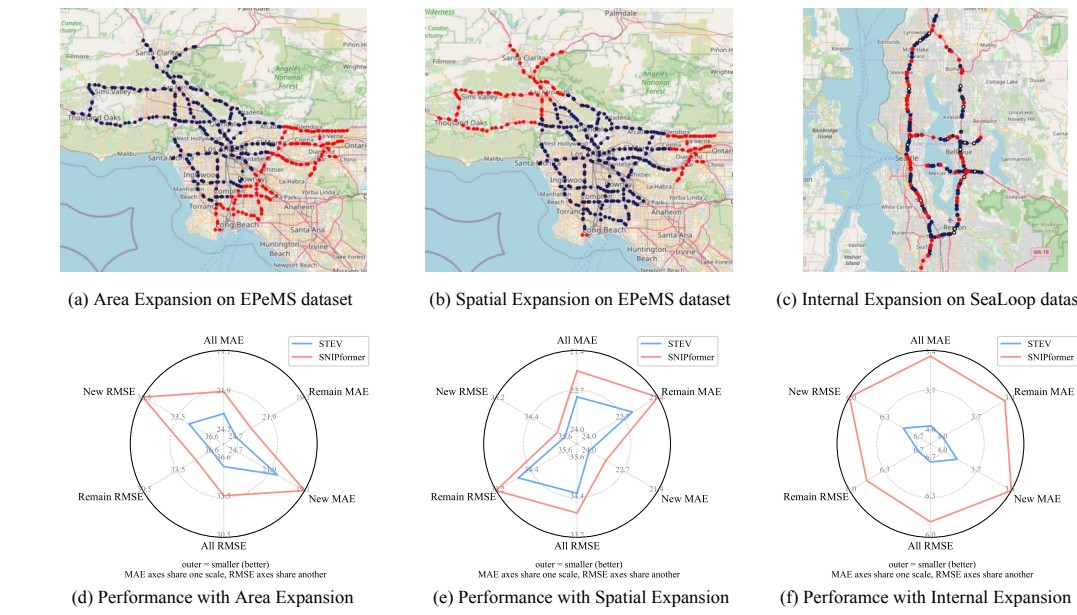

Figure 11: Visualization of geographic distribution and performance comparison with three different expansion scenarios. Red nodes denote the newly added nodes, dark-blue nodes indicate the remain nodes, and white nodes represent the retirements of remain nodes.

For each scenario, panels (d) to (f) report radar plots that summarize MAE and RMSE on All, Remain, and New nodes for the strongest baseline STEV and SNIPformer under the same forecasting setup. MAE related axes share a common scale, RMSE related axes share another, and smaller errors correspond to larger radii on the radar plots. Across the three expansion patterns, SNIPformer typically encloses a larger polygon than STEV, which indicates lower aggregate errors, while the relative gains on Remain and New nodes vary with the specific geographic expansion pattern. In some cases the improvements are more pronounced for newly added sensors in previously uncovered regions, whereas in others they are more balanced between Remain and New nodes. This suggests that the proposed priors and prompting mechanism can adapt to different spatial expansion regimes without introducing systematic degradation on either group of nodes. These gains hold under area, spatial, and internal expansion show that SNIPformer is robust to different geographic expansion modes.

## C    USE OF LLMs

In this work, we used large language models solely for polishing grammar and improving clarity. All research ideas, methodologies, experiments, analyses, and conclusions were independently conceived and conducted by the authors. The LLM was not used for generating research content, experiments, results, or references.

Table 9: Full comparison of the expanding-node forecasting results of different methods and SNIP-former. MAPE values are scaled by 100 for presentation.

| Model | Metric | EPeMS | | | PEMS04 | | | SeaLoop | | | NREL-AL | | |
|---|---|---|---|---|---|---|---|---|---|---|---|---|---|
| | | All | Remain | New | All | Remain | New | All | Remain | New | All | Remain | New |
| DLinear | MAE | 32.70 | 32.26 | 33.56 | 28.97 | 28.91 | 29.17 | 4.59 | 4.62 | 4.49 | 2.54 | 2.59 | 2.41 |
| | MAPE | 14.15 | 15.00 | 12.47 | 19.48 | 19.35 | 19.90 | 14.17 | 14.33 | 13.61 | 110.29 | 110.64 | 109.29 |
| | RMSE | 48.32 | 48.14 | 48.66 | 44.55 | 44.29 | 45.42 | 7.99 | 8.02 | 7.92 | 3.91 | 4.00 | 3.63 |
| | MRE | 0.10 | 0.10 | 0.10 | 0.13 | 0.13 | 0.13 | 0.08 | 0.09 | 0.08 | 0.21 | 0.21 | 0.21 |
| iTransformer | MAE | 26.83 | 26.65 | 27.16 | 24.76 | 24.74 | 24.83 | 4.29 | 4.31 | 4.21 | 1.94 | 1.97 | 1.83 |
| | MAPE | 10.91 | 11.39 | 9.96 | 16.08 | 16.02 | 16.29 | 12.71 | 12.80 | 12.40 | 100.71 | 101.10 | 99.60 |
| | RMSE | 41.40 | 41.22 | 41.73 | 39.62 | 39.34 | 40.52 | 7.54 | 7.56 | 7.46 | 3.36 | 3.44 | 3.12 |
| | MRE | 0.08 | 0.09 | 0.08 | 0.11 | 0.11 | 0.11 | 0.08 | 0.08 | 0.08 | 0.22 | 0.22 | 0.22 |
| DUET | MAE | 25.25 | 25.17 | 25.39 | 23.21 | 23.21 | 23.24 | 4.02 | 4.04 | 3.93 | 1.82 | 1.85 | 1.72 |
| | MAPE | 10.28 | 10.78 | 9.31 | 15.27 | 15.23 | 15.42 | 12.32 | 12.43 | 11.96 | 93.49 | 93.80 | 92.63 |
| | RMSE | 38.05 | 38.18 | 37.77 | 36.54 | 36.32 | 37.26 | 7.02 | 7.04 | 6.93 | 3.10 | 3.17 | 2.88 |
| | MRE | 0.08 | 0.08 | 0.07 | 0.10 | 0.10 | 0.10 | 0.07 | 0.07 | 0.07 | 0.21 | 0.21 | 0.21 |
| GWNET | MAE | 23.73 | 23.11 | 24.93 | 22.99 | 23.05 | 22.79 | 3.94 | 3.97 | 3.84 | 1.79 | 1.83 | 1.69 |
| | MAPE | 9.47 | 9.72 | 8.98 | 14.79 | 14.77 | 14.88 | 11.86 | 12.01 | 11.37 | 92.04 | 93.47 | 88.02 |
| | RMSE | 35.81 | 35.27 | 36.84 | 36.70 | 36.57 | 37.16 | 6.82 | 6.86 | 6.68 | 3.16 | 3.24 | 2.92 |
| | MRE | 0.07 | 0.07 | 0.07 | 0.10 | 0.10 | 0.10 | 0.07 | 0.07 | 0.07 | 0.23 | 0.23 | 0.23 |
| STID | MAE | 24.40 | 24.31 | 24.56 | 22.49 | 22.56 | 22.25 | 4.10 | 4.12 | 4.03 | 2.00 | 2.03 | 1.89 |
| | MAPE | 9.92 | 10.43 | 8.92 | 14.67 | 14.67 | 14.67 | 13.66 | 13.77 | 13.29 | 104.42 | 104.96 | 102.91 |
| | RMSE | 37.38 | 37.44 | 37.23 | 35.92 | 35.77 | 36.42 | 7.26 | 7.28 | 7.20 | 3.25 | 3.33 | 3.01 |
| | MRE | 0.08 | 0.08 | 0.07 | 0.10 | 0.10 | 0.10 | 0.08 | 0.08 | 0.07 | 0.17 | 0.17 | 0.17 |
| STAEformer | MAE | 24.86 | 24.66 | 25.27 | 22.95 | 23.03 | 22.67 | 4.15 | 4.17 | 4.09 | 1.91 | 1.95 | 1.81 |
| | MAPE | 9.94 | 10.38 | 9.07 | 14.76 | 14.71 | 14.91 | 13.08 | 13.15 | 12.83 | 87.52 | 87.75 | 86.86 |
| | RMSE | 38.34 | 38.26 | 38.50 | 36.75 | 36.63 | 37.14 | 7.38 | 7.39 | 7.37 | 3.29 | 3.37 | 3.05 |
| | MRE | 0.08 | 0.08 | 0.07 | 0.10 | 0.10 | 0.10 | 0.08 | 0.08 | 0.07 | 0.17 | 0.17 | 0.17 |
| STOP | MAE | 24.45 | 24.47 | 24.41 | 22.54 | 22.56 | 22.46 | 4.12 | 4.13 | 4.08 | 2.01 | 2.05 | 1.89 |
| | MAPE | 10.00 | 10.57 | 8.89 | 14.81 | 14.78 | 14.90 | 13.37 | 13.37 | 13.35 | 89.90 | 90.82 | 87.30 |
| | RMSE | 37.24 | 37.41 | 36.89 | 35.74 | 35.52 | 36.45 | 7.32 | 7.32 | 7.35 | 3.25 | 3.32 | 3.02 |
| | MRE | 0.08 | 0.08 | 0.07 | 0.10 | 0.10 | 0.10 | 0.08 | 0.08 | 0.07 | 0.16 | 0.16 | 0.16 |
| STKEC | MAE | 29.99 | 29.78 | 30.40 | 25.64 | 25.84 | 24.87 | 5.00 | 5.01 | 4.98 | 2.33 | 2.34 | 2.28 |
| | MAPE | 14.37 | 15.83 | 11.52 | 17.61 | 17.39 | 18.42 | 17.76 | 17.59 | 18.44 | 121.15 | 121.70 | 119.56 |
| | RMSE | 42.91 | 43.05 | 42.64 | 39.55 | 39.74 | 38.73 | 8.14 | 8.13 | 8.16 | 3.62 | 3.66 | 3.51 |
| | MRE | 0.09 | 0.10 | 0.09 | 0.12 | 0.12 | 0.12 | 0.09 | 0.09 | 0.09 | 0.21 | 0.21 | 0.21 |
| EAC | MAE | 28.74 | 28.23 | 29.75 | 24.05 | 24.27 | 23.21 | 4.72 | 4.73 | 4.72 | 2.16 | 2.17 | 2.14 |
| | MAPE | 12.24 | 12.83 | 11.06 | 18.14 | 17.74 | 19.58 | 17.00 | 16.62 | 18.43 | 114.49 | 115.24 | 112.29 |
| | RMSE | 40.33 | 39.80 | 41.35 | 36.51 | 36.79 | 35.41 | 7.82 | 7.81 | 7.86 | 3.37 | 3.39 | 3.32 |
| | MRE | 0.09 | 0.09 | 0.09 | 0.11 | 0.11 | 0.11 | 0.09 | 0.09 | 0.09 | 0.20 | 0.20 | 0.20 |
| STEV | MAE | 22.90 | 22.35 | 23.97 | 20.55 | 20.42 | 21.01 | 3.92 | 3.95 | 3.84 | 1.57 | 1.58 | 1.53 |
| | MAPE | 9.45 | 9.81 | 8.75 | 14.77 | 14.65 | 15.18 | 12.56 | 12.71 | 12.10 | 67.52 | 68.33 | 65.21 |
| | RMSE | 34.51 | 33.95 | 35.60 | 32.46 | 32.13 | 33.53 | 6.62 | 6.66 | 6.51 | 2.88 | 2.93 | 2.73 |
| | MRE | 0.07 | 0.07 | 0.07 | 0.09 | 0.09 | 0.09 | 0.07 | 0.07 | 0.07 | 0.15 | 0.14 | 0.15 |
| SNIPformer (ours) | MAE | 22.05 | 21.39 | 23.35 | 19.20 | 19.22 | 19.10 | 3.46 | 3.47 | 3.42 | 1.62 | 1.65 | 1.55 |
| | MAPE | 8.95 | 9.20 | 8.46 | 12.68 | 12.64 | 12.81 | 10.50 | 10.62 | 10.10 | 88.75 | 90.38 | 84.13 |
| | RMSE | 33.91 | 33.16 | 35.33 | 31.02 | 30.87 | 31.54 | 6.10 | 6.14 | 5.97 | 2.87 | 2.92 | 2.71 |
| | MRE | 0.07 | 0.07 | 0.07 | 0.09 | 0.09 | 0.09 | 0.06 | 0.06 | 0.06 | 0.15 | 0.15 | 0.15 |

Table 10: Full forecasting MAE results of different backbones with and without prompting modules. '-' denotes the method is invalid on the expansion scenario.

| Family | Model | EPeMS | | | PEMS04 | | | SeaLoop | | | NREL-AL | | |
|---|---|---|---|---|---|---|---|---|---|---|---|---|---|
| | | All | Remain | New | All | Remain | New | All | Remain | New | All | Remain | New |
| MLP based | DLinear | 32.70 | 32.26 | 33.56 | 28.97 | 28.91 | 29.17 | 4.59 | 4.62 | 4.49 | 2.54 | 2.59 | 2.41 |
| | +AttP | 32.37 | 31.91 | 33.25 | 28.58 | 28.52 | 28.79 | 4.58 | 4.61 | 4.48 | 2.50 | 2.55 | 2.37 |
| | +SNIP | 29.13 | 28.95 | 29.47 | 26.45 | 26.35 | 26.78 | 4.52 | 4.55 | 4.43 | 2.19 | 2.23 | 2.08 |
| | STID† | 24.40 | 24.31 | 24.56 | 22.49 | 22.56 | 22.25 | 4.10 | 4.12 | 4.03 | 2.00 | 2.03 | 1.89 |
| | +AttP | 24.35 | 24.26 | 24.53 | 22.48 | 22.55 | 22.23 | 4.11 | 4.13 | 4.04 | 2.01 | 2.05 | 1.90 |
| | +SNIP | 21.84 | 21.13 | 23.23 | 19.19 | 19.21 | 19.10 | 3.74 | 3.74 | 3.75 | 1.86 | 1.89 | 1.77 |
| Graph based | DCRNN | 25.09 | 24.55 | 26.14 | 23.17 | 23.21 | 23.04 | 4.08 | 4.11 | 4.01 | 1.93 | 1.96 | 1.83 |
| | +AttP | 25.05 | 24.45 | 26.23 | 22.83 | 22.89 | 22.64 | 4.10 | 4.14 | 3.99 | 1.94 | 1.97 | 1.84 |
| | +SNIP | 23.88 | 23.07 | 25.47 | 19.73 | 19.71 | 19.79 | 3.82 | 3.84 | 3.75 | 1.89 | 1.93 | 1.78 |
| | GWN† | 23.73 | 23.11 | 24.93 | 22.99 | 23.05 | 22.79 | 3.94 | 3.97 | 3.84 | 1.79 | 1.83 | 1.69 |
| | +AttP | 23.75 | 23.13 | 24.96 | 23.01 | 23.07 | 22.80 | 3.94 | 3.97 | 3.84 | 1.79 | 1.82 | 1.69 |
| | +SNIP | 23.41 | 22.79 | 24.62 | 19.93 | 20.11 | 19.31 | 3.80 | 3.83 | 3.71 | 1.77 | 1.81 | 1.66 |
| | GMAN | 26.06 | 25.75 | 26.68 | 21.77 | 22.07 | 20.78 | 4.20 | 4.24 | 4.08 | 2.91 | 2.96 | 2.78 |
| | +AttP | 31.27 | 30.92 | 31.94 | 21.90 | 22.10 | 21.23 | 4.20 | 4.24 | 4.09 | 2.68 | 2.74 | 2.51 |
| | +SNIP | 25.71 | 25.42 | 26.28 | 21.25 | 21.42 | 20.65 | 4.18 | 4.22 | 4.07 | 2.53 | 2.56 | 2.43 |
| Attention based | iTransformer | 26.83 | 26.65 | 27.16 | 24.76 | 24.74 | 24.83 | 4.29 | 4.31 | 4.21 | 1.94 | 1.97 | 1.83 |
| | +AttP | 26.81 | 26.64 | 27.14 | 24.76 | 24.74 | 24.84 | 4.29 | 4.31 | 4.21 | 1.95 | 1.99 | 1.84 |
| | +SNIP | 24.67 | 24.14 | 25.71 | 21.55 | 21.57 | 21.50 | 4.02 | 4.02 | 4.01 | 1.84 | 1.88 | 1.74 |
| | DUET | 25.25 | 25.17 | 25.39 | 23.21 | 23.21 | 23.24 | 4.02 | 4.04 | 3.93 | 1.82 | 1.85 | 1.72 |
| | +AttP | 25.18 | 25.10 | 25.32 | 23.08 | 23.08 | 23.08 | 4.02 | 4.04 | 3.93 | 1.82 | 1.86 | 1.72 |
| | +SNIP | 23.16 | 22.56 | 24.34 | 20.05 | 19.92 | 20.48 | 3.60 | 3.58 | 3.67 | 1.74 | 1.77 | 1.66 |
| | STAEformer† | 24.86 | 24.66 | 25.27 | 22.95 | 23.03 | 22.67 | 4.15 | 4.17 | 4.09 | 1.95 | 1.99 | 1.84 |
| | +AttP | 24.80 | 24.62 | 25.15 | 22.92 | 23.00 | 22.64 | 4.19 | 4.21 | 4.12 | 1.93 | 1.96 | 1.81 |
| | +SNIP | 23.75 | 23.05 | 25.13 | 21.43 | 21.55 | 21.01 | 3.83 | 3.86 | 3.73 | 1.90 | 1.94 | 1.80 |
| Hybrid Architecture | TESTAM† | - | - | - | - | - | - | - | - | - | - | - | - |
| | +AttP | 29.51 | 29.59 | 29.36 | 26.75 | 26.74 | 26.75 | 4.04 | 4.06 | 3.95 | 2.42 | 2.46 | 2.30 |
| | +SNIP | 26.22 | 25.70 | 27.26 | 24.45 | 24.16 | 25.42 | 3.68 | 3.68 | 3.71 | 1.88 | 1.92 | 1.78 |

Table 11: Full forecasting RMSE results of different backbones with and without prompting modules. '-' denotes the method is invalid on the expansion scenario.

| Family | Model | EPeMS | | | PEMS04 | | | SeaLoop | | | NREL-AL | | |
|---|---|---|---|---|---|---|---|---|---|---|---|---|---|
| | | All | Remain | New | All | Remain | New | All | Remain | New | All | Remain | New |
| MLP based | DLinear | 48.32 | 48.14 | 48.66 | 44.55 | 44.29 | 45.42 | 7.99 | 8.02 | 7.92 | 3.91 | 4.00 | 3.63 |
| | +AttP | 47.49 | 47.22 | 48.00 | 43.61 | 43.32 | 44.57 | 7.94 | 7.96 | 7.87 | 3.88 | 3.97 | 3.60 |
| | +SNIP | 43.30 | 43.41 | 43.08 | 41.09 | 40.78 | 42.12 | 7.82 | 7.84 | 7.74 | 3.59 | 3.68 | 3.34 |
| | STID† | 37.38 | 37.44 | 37.23 | 35.92 | 35.77 | 36.42 | 7.26 | 7.28 | 7.20 | 3.25 | 3.33 | 3.01 |
| | +AttP | 37.34 | 37.39 | 37.22 | 35.94 | 35.79 | 36.43 | 7.28 | 7.29 | 7.22 | 3.27 | 3.36 | 3.03 |
| | +SNIP | 33.73 | 32.90 | 35.31 | 31.03 | 30.86 | 31.63 | 6.76 | 6.76 | 6.77 | 3.07 | 3.13 | 2.88 |
| Graph based | DCRNN | 37.45 | 37.04 | 38.23 | 36.75 | 36.60 | 37.26 | 7.13 | 7.15 | 7.06 | 3.40 | 3.48 | 3.15 |
| | +AttP | 37.25 | 36.76 | 38.20 | 36.25 | 36.14 | 36.60 | 7.23 | 7.27 | 7.07 | 3.41 | 3.49 | 3.16 |
| | +SNIP | 35.70 | 34.86 | 37.31 | 31.53 | 31.34 | 32.17 | 6.78 | 6.81 | 6.66 | 3.31 | 3.40 | 3.06 |
| | GWN† | 35.81 | 35.27 | 36.84 | 36.70 | 36.57 | 37.16 | 6.82 | 6.86 | 6.68 | 3.16 | 3.24 | 2.92 |
| | +AttP | 35.82 | 35.26 | 36.88 | 36.73 | 36.59 | 37.19 | 6.80 | 6.83 | 6.68 | 3.16 | 3.24 | 2.92 |
| | +SNIP | 35.54 | 34.97 | 36.63 | 32.13 | 32.25 | 31.71 | 6.69 | 6.73 | 6.55 | 3.13 | 3.20 | 2.90 |
| | GMAN | 38.87 | 38.76 | 39.08 | 35.21 | 35.46 | 34.36 | 7.64 | 7.69 | 7.49 | 4.27 | 4.36 | 4.01 |
| | +AttP | 45.46 | 45.58 | 45.22 | 35.58 | 35.67 | 35.27 | 7.64 | 7.69 | 7.47 | 4.09 | 4.19 | 3.78 |
| | +SNIP | 38.52 | 38.45 | 38.65 | 34.49 | 34.57 | 34.20 | 7.58 | 7.62 | 7.41 | 3.91 | 3.99 | 3.67 |
| Attention based | iTransformer | 41.40 | 41.22 | 41.73 | 39.62 | 39.34 | 40.52 | 7.54 | 7.56 | 7.46 | 3.36 | 3.44 | 3.12 |
| | +AttP | 41.38 | 41.21 | 41.71 | 39.62 | 39.35 | 40.53 | 7.54 | 7.57 | 7.46 | 3.36 | 3.45 | 3.11 |
| | +SNIP | 38.14 | 37.68 | 39.02 | 34.55 | 34.42 | 34.97 | 7.15 | 7.18 | 7.05 | 3.23 | 3.31 | 2.99 |
| | DUET | 38.05 | 38.18 | 37.77 | 36.54 | 36.32 | 37.26 | 7.02 | 7.04 | 6.93 | 3.10 | 3.17 | 2.88 |
| | +AttP | 37.97 | 38.11 | 37.68 | 36.35 | 36.15 | 37.02 | 7.03 | 7.05 | 6.94 | 3.09 | 3.17 | 2.86 |
| | +SNIP | 35.00 | 34.33 | 36.28 | 31.72 | 31.40 | 32.79 | 6.17 | 6.12 | 6.33 | 2.98 | 3.05 | 2.77 |
| | STAEformer† | 38.34 | 38.26 | 38.50 | 36.75 | 36.63 | 37.14 | 7.38 | 7.39 | 7.37 | 3.33 | 3.42 | 3.07 |
| | +AttP | 38.19 | 38.15 | 38.25 | 36.72 | 36.60 | 37.11 | 7.44 | 7.45 | 7.40 | 3.30 | 3.39 | 3.04 |
| | +SNIP | 36.55 | 35.74 | 38.08 | 34.67 | 34.82 | 34.18 | 6.99 | 7.06 | 6.78 | 3.21 | 3.29 | 2.97 |
| Hybrid Architecture | TESTAM† | - | - | - | - | - | - | - | - | - | - | - | - |
| | +AttP | 41.74 | 41.87 | 41.48 | 38.83 | 38.71 | 39.22 | 7.08 | 7.11 | 6.99 | 3.63 | 3.72 | 3.38 |
| | +SNIP | 38.25 | 37.54 | 39.61 | 35.03 | 34.56 | 36.56 | 6.42 | 6.41 | 6.44 | 3.10 | 3.17 | 2.91 |

