# OpenReview forum: "Towards Expanding-Node Spatial-Temporal Forecasting: A Structured Node Interaction Prompting Perspective"
_ICLR.cc/2026/Conference — Submitted to ICLR 2026_

### Official Review · Reviewer_k2Rf · 2025-10-25

**Soundness:** 3
**Presentation:** 3
**Contribution:** 2
**Rating:** 4
**Confidence:** 4

**Summary:**

This paper addresses the problem of spatial-temporal forecasting with expanding nodes in a flexible and efficient manner. They propose SNIP, a method that constructs structured static priors while refining them with dynamic information. The paper is well-motivated and clearly organized. Experiments on four datasets demonstrate consistent improvements over baseline methods, with enhanced scalability and reduced retraining costs.

**Strengths:**

1. The paper addresses the challenging problem of expanding-node spatiotemporal forecasting, leveraging priors from historical observations without relying on learnable node embeddings.
2. The proposed method decouples prior information into static and dynamic components, adapting the dynamic prior to the expansion stage in a model-agnostic manner.
3.  The paper is well-structured and easy to understand.

**Weaknesses:**

1. The method is presented as model-agnostic but is evaluated on limited STF architectures. Validation on diverse backbones (e.g., graph-, attention-, and convolution-based) is needed to support this claim.
2.  The current datasets involve relatively small node scales and fixed four-stage splits. To better assess scalability and adaptability, experiments should include larger and more dynamic streams, such as _Air-Stream_ with thousands of nodes and _PEMS-Stream_ with multiple expansion phases (655 → 715 → 786 → 822 → 834 → 850 → 871).
3. The paper provides no rationale or empirical analysis for critical hyperparameters (kpca, ktopo, kdelay, kcorr ) or for the fixed node ratio (80% observed, 20% new, 5% deleted). Comprehensive sensitivity studies are needed to assess how these design choices affect model performance and robustness.

**Questions:**

1.  The description in Appendix A.3 is brief, more architectural details and a clearer diagram or pseudocode would improve reproducibility.
2.  As the method is claimed to be model-agnostic, have the authors tested it on other STF backbones? Results on more diverse architectures and larger, evolving datasets (e.g., _Air-Stream_ or _PEMS-Stream_ with multiple expansion phases) would better demonstrate scalability.
3.  Important hyperparameters (e.g., _kpca_, _n_, _ktopo_, _kdelay_, _kcorr_) and node ratios (80% observed, 20% new, 5% deleted) are fixed without justification.
4.  The dataset name appears inconsistently as “NERL-AL” and “NREL-AL.” Please clarify the correct form and ensure consistency.

---

> ### Author Response · Authors · 2025-11-22
> **The First Part of the Response to Reviewer k2Rf**
>
> **Response to Reviewer** **k2Rf:**
>
> Dear Reviewer k2Rf:
>
> We sincerely thank you for taking the time and effort to provide valuable feedback on our paper. We considered all remarks with care and address them one by one, noting the changes made in the manuscript.
>
> ---
>
> > **W1: The method is presented as model-agnostic but is evaluated on limited STF architectures. Validation on diverse backbones (e.g., graph-, attention-, and convolution-based) is needed to support this claim.**
>
> Thank you for pointing out the need to substantiate the model-agnostic claim with broader backbone coverage. **We have extended the evaluation to four families of STF or TF architectures**, and report results after inserting SNIP as a drop-in replacement for node-specific learnable embeddings.
>
> The four families are: **MLP based** (DLinear[1], STID[2]), **graph based** (DCRNN[3], GWNET[4], GMAN[5]), **attention based** (iTransformer[6], DUET[7], STAEformer[8]), and a **hybrid architecture** (TESTAM[9], three experts for unique spatio-temporal modeling). SNIP is designed to remove node tied parameters, so *a backbone is considered compatible only if it can operate in the expanding node setting after those parameters are removed*.
>
> We summarize MAE on EPeMS and NREL-AL below, the dash ‘-’ in the table indicates that the backbone becomes inoperable under expansion once node specific parameters are removed. Full results for all datasets are provided in the revised Appendix B.4 (Table 10 and Table 11).
>
> |||EPeMS|||NREL-AL||
> |------------|-----|------|-----|----|-------|----|
> ||All|Remain|New|All|Remain|New|
> |**DLinear**|32.70|32.26|33.56|2.54|2.59|2.41|
> |+AttP|32.37|31.91|33.25|2.50|2.55|2.37|
> |+SNIP|29.13|28.95|29.47|2.19|2.23|2.08|
> |**STID†**|24.40|24.31|24.56|2.00|2.03|1.89|
> |+AttP|24.35|24.26|24.53|2.01|2.05|1.90|
> |+SNIP|21.84|21.13|23.23|1.86|1.89|1.77|
> |**DCRNN**|25.09|24.55|26.14|1.93|1.96|1.83|
> |+AttP|25.05|24.45|26.23|1.94|1.97|1.84|
> |+SNIP|23.88|23.07|25.47|1.89|1.93|1.78|
> |**GWNET†**|23.73|23.11|24.93|1.79|1.83|1.69|
> |+AttP|23.75|23.13|24.96|1.79|1.82|1.69|
> |+SNIP|23.41|22.79|24.62|1.77|1.81|1.66|
> |**GMAN**|26.06|25.75|26.68|2.91|2.96|2.78|
> |+AttP|31.27|30.92|31.94|2.68|2.74|2.51|
> |+SNIP|25.71|25.42|26.28|2.53|2.56|2.43|
> |**iTransformer**|26.83|26.65|27.16|1.94|1.97|1.83|
> |+AttP|26.81|26.64|27.14|1.95|1.99|1.84|
> |+SNIP|24.67|24.14|25.71|1.84|1.88|1.74|
> |**DUET**|25.25|25.17|25.39|1.82|1.85|1.72|
> |+AttP|25.18|25.10|25.32|1.82|1.86|1.72|
> |+SNIP|23.16|22.56|24.34|1.74|1.77|1.66|
> |**STAEformer†**|24.86|24.66|25.27|1.95|1.99|1.84|
> |+AttP|24.80|24.62|25.15|1.93|1.96|1.81|
> |+SNIP|23.75|23.05|25.13|1.90|1.94|1.80|
> |**TESTAM†**|-|-|-|-|-|-|
> |+AttP|29.51|29.59|29.36|2.42|2.46|2.30|
> |+SNIP|26.22|25.70|27.26|1.88|1.92|1.78|
>
>
>
> Across MLP, graph, attention, and hybrid families, **inserting SNIP yields consistent gains on All, Remain, and New nodes**, which supports the claim that SNIP is a model-agnostic prompting layer rather than a backbone specific trick. Compared with the attention-based learning Prompting strategy (AttP), our method establishes valuable priors and, through flexible learned fine-tuning, **achieves more beneficial node-discriminative prompting along with superior prediction outcomes.**
>
> We appreciate the suggestion and have incorporated these broader results and implementation notes into the revised paper.
>
> [1] A. Zeng et al. Are Transformers Effective for Time Series Forecasting? AAAI, 2023.
>
> [2] Z. Shao et al. Spatial-Temporal Identity: A Simple yet Effective Baseline for Multivariate Time Series Forecasting. CIKM, 2022.
>
> [3] Y. Li et al. Diffusion Convolutional Recurrent Neural Network: Data-Driven Traffic Forecasting. ICLR, 2018.
>
> [4] Z. Wu et al. Graph WaveNet for Deep Spatial-Temporal Graph Modeling. IJCAI, 2019.
>
> [5] C. Zheng et al. GMAN, A Graph Multi-Attention Network for Traffic Prediction. AAAI, 2020.
>
> [6] B. Liu et al. iTransformer: Inverted Transformers Are Effective for Time Series Forecasting. ICLR, 2024.
>
> [7] X. Qiu et al. DUET: Dual Clustering Enhanced Multivariate Time Series Forecasting. KDD, 2025.
>
> [8] H. Liu et al. STAEformer: Spatio-Temporal Adaptive Embedding Makes Vanilla Transformer SOTA for Traffic Forecasting. CIKM, 2023.
>
> [9] H. Lee et al. TESTAM, A Time-Enhanced Spatio-Temporal Attention Model with Mixture of Experts. ICLR, 2024.

---

> ### Author Response · Authors · 2025-11-22
> **The Second Part of the Response to Reviewer k2Rf**
>
> > **W2. The current datasets involve relatively small node scales and fixed four-stage splits. To better assess scalability and adaptability, experiments should include larger and more dynamic streams, such as Air-Stream with thousands of nodes and PEMS-Stream with multiple expansion phases (655 → 715 → 786 → 822 → 834 → 850 → 871).**
>
> Thank you for the helpful suggestion. We agree that validating scalability under larger node counts and more dynamic streams is important. Our problem definition **abstracts each expansion into a before stage and an after stage with scarce data for newly added nodes**, so **a multi phase stream can be handled as a sequence of such expansions with fine tuning at each phase**. This aligns with the continual or streaming setting where sensors arrive over time, as discussed in recent continual ST forecasting work such as EAC, and in streaming benchmarks like PEMS-Stream and Air-Stream.
>
> To directly address the request, we added experiments on two standard streaming benchmarks. PEMS-Stream contains multiple expansion phases with node counts 655 to 871 over 2011 to 2017, and Air-Stream spans 2016 to 2019 with more than one thousand sensors, both designed for continual evaluation across phases.  We follow the EAC protocol for phase wise evaluation and compare against strong continual baselines EAC and STKEC.
>
> Below we report MAE and RMSE on 12-steps ahead prediction for SNIPformer and a lightweight backbone with SNIP (STID†+SNIP). Full results and scripts are included in the revised Appendix B.5 (Table 6 and Table 7).
>
> - PEMS-Strem
>
>
> |ModelonPEMS-Stream|Metric|2011|2012|2013|2014|2015|2016|2017|
> |--------------------|------|--------:|--------:|--------:|--------:|--------:|--------:|--------:|
> |STKEC|MAE|15.80|15.77|15.86|16.77|16.27|15.64|17.16|
> ||RMSE|24.63|25.00|25.96|27.60|26.85|27.91|28.17|
> |EAC|MAE|14.51|14.23|14.37|15.20|14.87|14.28|15.91|
> ||RMSE|22.22|22.14|23.13|24.31|24.26|25.65|25.87|
> |SNIPformer|MAE|**11.79**|**11.20**|**11.20**|*11.94*|*11.58*|*11.07*|*12.76*|
> ||RMSE|**18.02**|**17.73**|*18.47*|**19.24**|**19.37**|*21.51*|*21.55*|
> |STID†+SNIP|MAE|*12.30*|*11.29*|*11.23*|11.59|11.51|**10.86**|**12.62**|
> ||RMSE|*19.20*|*18.34*|**18.18**|*19.40*|*19.58*|**21.23**|**21.18**|
>
>
>
> - Air-Stream
>
>
> |ModelonAir-Stream|Metric|2016|2017|2018|2019|
> |-------------------|------|--------:|--------:|--------:|--------:|
> |STKEC|MAE|31.04|27.04|*20.16*|21.46|
> ||RMSE|49.91|41.75|*34.08*|33.28|
> |EAC|MAE|31.39|25.75|20.71|21.25|
> ||RMSE|49.99|39.34|34.20|32.94|
> |SNIPformer|MAE|*25.55*|*23.38*|20.56|*19.19*|
> ||RMSE|*41.31*|*36.80*|35.35|*30.50*|
> |STID†+SNIP|MAE|**24.55**|**21.92**|**19.00**|**18.84**|
> ||RMSE|**40.19**|**35.17**|**32.47**|**30.26**|
>
>
>
> Results show that both SNIPformer and the lightweight STID†+SNIP consistently outperform continual baselines across multiple phases on PEMS-Stream and Air-Stream, which supports scalability and adaptability in longer and larger streams.
>
> We appreciate the suggestion. We have added these results and implementation details to the revised Appendix B.5 (Table 6 and Table 7).

---

> ### Author Response · Authors · 2025-11-22
> **The Third Part of the Response to Reviewer k2Rf**
>
> > **W3. The paper provides no rationale or empirical analysis for critical hyperparameters (kpca, ktopo, kdelay, kcorr) or for the fixed node ratio (80\% observed, 20\% new, 5\% deleted). Comprehensive sensitivity studies are needed to assess how these design choices affect model performance and robustness.**
>
> Thank you for highlighting the need to justify and analyze our hyperparameters and the fixed node ratios. We apologize for the omission of this analysis in the appendix of the previous version of the manuscript. We now provide a consolidated rationale and sensitivity study. For each group of choices we performed a grid search and selected settings by jointly considering predictive accuracy and training or preprocessing overhead.
>
> **Embedding dimensions for interaction features: $k_\text{topo}, k_\text{delay}, k_\text{corr}$.**
> These dimensions control the low dimensional representations used by the topology prior, the optimal time delay, and the correlation strength. To keep the three subspaces balanced when concatenated, we tied them to the same value and swept a wide range.
>
>
> |$k_\text{topo},k_\text{delay},k_\text{corr}$|4|8|16|24|32|40|48|56|64|
> |---------------------|----:|----:|----:|----:|----:|----:|----:|----:|----:|
> |MAE|19.36|19.20|19.20|19.29|19.16|19.14|19.20|19.26|19.41|
> |RMSE|31.22|31.02|31.04|31.17|30.94|31.03|31.07|31.14|31.27|
>
>
>
> The metric exhibits minimal fluctuation across all tested configurations, consistently outperforming the optimal baseline method, while larger dimensions increase both prior computation and the cost of dynamic refinement. Therefore, we set $k_\text{topo}=k_\text{delay}=k_\text{corr}$=8 for a balance between predictive accuracy and training or preprocessing overhead.
>
> **Number of periodic PCA components**, $k_\text{pca}$.
> We choose $k_\text{pca}$ by jointly considering explained variance and end to end accuracy. Using principal components that capture the bulk of variance is a standard criterion in PCA based feature construction.
>
>
> |$k_\text{pca}$|4|8|16|24|32|40|48|56|64|
> |-----|----:|----:|----:|----:|----:|----:|----:|----:|----:|
> |MAE|19.44|19.24|19.27|19.20|19.26|19.23|19.40|19.25|19.34|
> |RMSE|31.34|31.01|31.13|31.02|31.20|31.07|31.31|31.26|31.31|
>
> We observe a flat region, and pick $k_\text{pca} = 24$ to keep preprocessing cost low while retaining prediction accuracy.
>
> **Number of periodicities n.**
> We tested one daily cycle, one weekly cycle, and a two cycle combination.
>
>
> |number of periods|n=1 [day]|n=1 [week]|n=2 [day+week]|
> |-----------------|----------:|-----------:|---------------:|
> |MAE|19.36|19.40|19.20|
> |RMSE|31.08|31.27|31.02|
>
>
> Using both day and week yields the best accuracy, which is consistent with multi seasonal behavior commonly observed in traffic and energy series.
>
>
> **Hann window size for Welch method in the time-delayed interaction priors computation.**
>
> When constructing the time delayed interaction prior, we use Welch’s segment averaging with a Hann window. Longer windows reduce estimator variance and spectral leakage, at the cost of time resolution, while shorter windows improve time localization. We swept the window size and observed convergence to a stable band as the window grows. We set the Hann window size to $T=12$, which matches the forecasting horizon, providing a balanced compromise between variance reduction and time localization.
>
>
> |Hann window size|4|6|12|24|36|48|
> |----------------|----:|----:|----:|----:|----:|----:|
> |MAE|19.29|19.27|19.20|19.34|19.20|19.19|
> |RMSE|31.15|31.13|31.02|31.21|31.08|31.05|
>
>
>
> **Node ratio sensitivity.**
> We further varied the fraction of newly added nodes and the retirement rate of existing nodes to probe robustness under different expansion profiles.
>
> *Proportion of new nodes*
>
>
> |newadd rate|20\%|30\%|40\%|50\%|60\%|70\%|80\%|
> |-----------|----:|----:|----:|----:|----:|----:|----:|
> |MAE|19.20|19.07|19.34|19.32|19.27|19.45|19.34|
> |RMSE|31.02|30.86|31.23|31.23|31.14|31.35|31.07|
>
>
> *Retirement rate of remain nodes*
>
>
> |retire rate|5\%|15\%|25\%|35\%|45\%|55\%|
> |-----------|-----|-----|-----|-----|-----|-----|
> |MAE|19.20|19.37|18.98|19.04|18.62|18.64|
> |RMSE|31.02|31.23|30.52|30.80|30.17|30.30|
>
>
> Across a wide range of ratios, SNIPformer remains stable, which suggests that the priors help regularize the problem when the composition of nodes shifts.
>
> We have added full reasults on All, Remain, and New subsets to the experiment Section 5.3 and Appendix B.3. Thanks again for your suggestions.

---

> ### Author Response · Authors · 2025-11-22
> **The Fourth Part of the Response to Reviewer k2Rf**
>
> > **Q1:The description in Appendix A.3 is brief, more architectural details and a clearer diagram or pseudocode would improve reproducibility.**
>
> Thank you for the suggestion. We have expanded Appendix A.3 to include a detailed architectural diagram of SNIPformer, along with pseudocode for prior construction, pre-training, and fine-tuning based on SNIP.
>
> ---
>
> > **Q2:As the method is claimed to be model-agnostic, have the authors tested it on other STF backbones? Results on more diverse architectures and larger, evolving datasets (e.g., Air-Stream or PEMS-Stream with multiple expansion phases) would better demonstrate scalability.**
>
> Please see our response to W1 and W2, where we add results on additional backbones and on PEMS-Stream and Air-Stream, together with implementation notes for operating in multi phase expansion settings.
>
> ---
>
> > **Q3:Important hyperparameters (e.g.,$k_{pca}$,$n$,$k_{topo}$,$k_{delay}$,$k_{corr}$) and node ratios (80\% observed, 20\% new, 5\% deleted) are fixed without justification.**
>
> Please see our response to W3, where we provide the grid ranges, selection rationale, and sensitivity analyses for $k_\text{pca}$, $n$, $k_\text{topo}$, $k_\text{delay}$, $k_\text{corr}$, as well as ablations over new node and retirement ratios.
>
> ---
>
> > **Q4:The dataset name appears inconsistently as “NERL-AL” and “NREL-AL.” Please clarify the correct form and ensure consistency.**
>
> Thank you for catching this. The correct form is NREL-AL. We have unified the name across the manuscript, figures, and appendix. We appreciate your careful reading and have updated the paper accordingly.
>
> ---
>
> We greatly appreciate your valuable feedback, and we will incorporate these detailed discussions into the final revision of the paper. We hope the responses above address your concerns, and **if possible, we kindly request that you reconsider increasing the score**. Should you have any further suggestions, we are more than happy to discuss them and make necessary improvements to the paper.

---

> > ### Comment · Reviewer_k2Rf · 2025-11-25
> >
> > Thanks for the response. I do not have further comments, I will adjust score accordingliy.

---

> > > ### Author Response · Authors · 2025-11-26
> > >
> > > Thank you for the follow up and for considering an adjusted score. We greatly appreciate your time and look forward to your further positive feedback in the discussion phase.

---

> > > ### Author Response · Authors · 2025-11-28
> > >
> > > Dear Reviewer k2Rf,
> > >
> > > Thank you for your thoughtful review and for raising the contribution score from 2 to 3. We appreciate the recognition.
> > >
> > > We believe the added experiments and clarifications, broader backbone coverage, multi stage evaluations, and hyperparameter sensitivity analyses, have addressed your concerns about generality and robustness. **If this aligns with your assessment, we would be grateful if you could consider increasing the overall score**. We greatly appreciate your time and effort in reviewing our work. Your suggestions have been very helpful and have strengthened the manuscript.
> > >
> > > Best regards,
> > >
> > > The Authors

---

### Official Review · Reviewer_MB12 · 2025-10-25

**Soundness:** 3
**Presentation:** 3
**Contribution:** 3
**Rating:** 6
**Confidence:** 4

**Summary:**

This paper “introduces SNIP, which is a model-agnostic framework that tackles the challenge of expanding-node spatial-temporal forecasting, where sensor networks evolve over time (e.g., new traffic sensors are added or removed). Traditional spatio-temporal models fail in such dynamic settings because their parameters are tied to fixed node sets. SNIP breaks this dependency by constructing static node priors from historical sequences and topology, including periodic, topological, and time-delayed interaction features, with PCA and spectral embeddings to capture heterogeneity and correlations without learnable node embeddings. These priors are then dynamically refined via diffusion graph convolutions and initialized for new nodes using similarity-weighted mixtures, enabling efficient few-shot adaptation. Experiments across four real-world datasets demonstrate that SNIP and its instantiation SNIPformer outperform strong baselines like STEV and continual-learning approaches.

**Strengths:**

S1. This paper proposes the study of expanding-node spatial-temporal forecasting, a realistic yet largely neglected setting where the number of nodes in a sensor network changes over time. By formally defining this problem and identifying its key challenges, the paper establishes a research direction that better reflects operational realities and fills an important gap in the spatio-temporal forecasting literature.

S2. One of the framework’s contributions lies in its ability to dynamically adapt to new nodes with minimal data. Through a diffusion-based graph convolution refinement process, SNIP continuously updates its priors as network conditions evolve. When new nodes are introduced, it uses a similarity-weighted initialization strategy that blends knowledge from existing nodes based on their correlation strength.

S3. Empirically, the proposed model demonstrates consistent improvements across a range of benchmark datasets and evaluation settings.

**Weaknesses:**

W1. The paper assumes a single expanding stage for sensor nodes, which simplifies the evolving network process. In real-world systems, however, sensor deployment and removal typically occur continuously over time rather than in one expansion phase. It remains unclear how the proposed framework would handle multi-stage or streaming expansions, where node sets evolve incrementally and model adaptation must occur online.

W2. The model incurs a non-trivial preprocessing cost to compute multi-cycle PCA, cross-spectral correlations, and eigen-decompositions for spectral embeddings, especially when the full historical records are utilized for calculation. These steps may scale poorly for large networks or high-frequency data streams.

W3. The paper proposes the Decomposition Hypothesis that optimal node promptings can be decomposed into static and dynamic components. However, this claim might be conceptually plausible and only empirically validated. There is no formal theoretical analysis when combining static priors with diffusion-based refinement.

W4.  The topology priors in SNIP are constructed directly from a fixed adjacency matrix, which may not accurately capture the true correlations or dynamic dependencies among nodes. Prior studies in spatial-temporal forecasting have shown that such predefined graphs can be suboptimal or even misleading.

**Questions:**

Q1. Line 201, $X_{:,:,1}$ appears to be inconsistent with the earlier definition of $X$, which is a 3-dimensional tensor.

Q2. The connection between the diffusion convolution operation and the prediction process during the expansion phase is not entirely clear. Is the diffusion convolution applied only during the stage of $\tau_{2}$ for training?

Q3. Figure 1(b) is not very intuitive or informative in illustrating the differences among methods. The authors may consider redesigning it with clearer visual cues or a more concrete comparative example to better highlight the distinctions.

---

> ### Author Response · Authors · 2025-11-22
> **The First Part of the Response to Reviewer MB12**
>
> **Response to Reviewer MB12:**
>
> Dear Reviewer MB12:
>
> We sincerely thank you for your thoughtful review and constructive suggestions, we have examined each comment carefully and provide a point by point response with the corresponding revisions.
>
> ---
>
> > **W1. The paper assumes a single expanding stage for sensor nodes, which simplifies the evolving network process. In real-world systems, however, sensor deployment and removal typically occur continuously over time rather than in one expansion phase. It remains unclear how the proposed framework would handle multi-stage or streaming expansions, where node sets evolve incrementally and model adaptation must occur online.**
>
>
> Thank you for the insightful comment. We agree that real systems often evolve continuously. In our problem definition, **each expansion is abstracted as a before stage and an after stage with scarce data for newly added nodes**. Under this abstraction, *a multi stage stream is a sequence of such expansions.*  After the base stage, SNIP constructs node priors once. At each subsequent expansion, priors for new nodes can be computed quickly from their short post expansion history, or formed by a similarity weighted mixture from existing nodes when reliable, then the model performs lightweight fine tuning on the short window.
>
> When latency is critical, inference can even proceed in a zero shot manner using the new priors alone, although accuracy may be impacted by distribution shift. This protocol aligns with the setting emphasized by STEV, new nodes have only short histories and timely prediction is required, while *our framework does not require joint access to all phases during training and adapts phase by phase.*
>
> To demonstrate behavior under multi phase streams, we follow the EAC protocol and evaluate 12 step forecasting on PEMS-Stream across seven yearly phases. Results are below. Full results are provided in Appendix B.5.
>
>
> |Model on PEMS-Stream|Metric|2011|2012|2013|2014|2015|2016|2017|
> |--------------------|------|--------:|--------:|--------:|--------:|--------:|--------:|--------:|
> |STKEC|MAE|15.80|15.77|15.86|16.77|16.27|15.64|17.16|
> ||RMSE|24.63|25.00|25.96|27.60|26.85|27.91|28.17|
> |EAC|MAE|14.51|14.23|14.37|15.20|14.87|14.28|15.91|
> ||RMSE|22.22|22.14|23.13|24.31|24.26|25.65|25.87|
> |SNIPformer|MAE|**11.79**|**11.20**|**11.20**|*11.94*|*11.58*|*11.07*|*12.76*|
> ||RMSE|**18.02**|**17.73**|*18.47*|**19.24**|**19.37**|*21.51*|*21.55*|
> |STID†+SNIP|MAE|*12.30*|*11.29*|*11.23*|11.59|11.51|**10.86**|**12.62**|
> ||RMSE|*19.20*|*18.34*|**18.18**|*19.40*|*19.58*|**21.23**|**21.18**|
>
>
> These results show that a lightweight MLP backbone, STID, maintains strong performance across multiple expansion phases once augmented with SNIP, and SNIPformer also performs consistently well. **Both findings support the adaptability and scalability of our framework in multi stage settings.**
>
> We acknowledge that constructing more benchmarks with fine grained phase annotations, and exploring fully online adaptation policies, are important next steps.

---

> ### Author Response · Authors · 2025-11-22
> **The Second Part of the Response to Reviewer MB12**
>
> > **W2: The model incurs a non-trivial preprocessing cost to compute multi-cycle PCA, cross-spectral correlations, and eigen-decompositions for spectral embeddings, especially when the full historical records are utilized for calculation. These steps may scale poorly for large networks or high-frequency data streams.**
>
> Thank you for raising the computational cost concern. The three preprocessing steps in SNIP, periodic PCA, time-delayed interaction estimation, and spectral embeddings, are performed offline, then cached and reused across backbones and across expansion phases. Training and inference only consume the concatenated prompts and the lightweight refinement module.
>
> Concretely, periodic segmentation reduces the sample size for each PCA, which enables parallel execution. For the time-delayed interaction prior, Welch CSD is computed with zero overlap, which further lowers runtime. On EPeMS, measured on the same hardware and implementation stack, the preprocessing time is `SNIP 2.61 minutes` (for three priors construction), `STEV 0.21 minutes`(for series augmentation), `GMAN 252 minutes`(for Spatial Embedding construction based on random walk). With optimization of modern libraries, the overall cost is acceptable.
>
> We acknowledge that larger networks and higher sampling rates demand additional care. We plan to continue optimizing the prior construction pipeline, for example through parallelization, incremental factorization, and cached updates for only the newly added nodes, so that SNIP remains scalable on large-scale streams.
>
> ---
>
> > **W3: The paper proposes the Decomposition Hypothesis that optimal node promptings can be decomposed into static and dynamic components. However, this claim might be conceptually plausible and only empirically validated. There is no formal theoretical analysis when combining static priors with diffusion-based refinement.**
>
> Thank you for highlighting the need for a clearer theoretical footing. Our hypothesis models a node’s prompting additively as $p_i(t) = q_i + r_i(t)$, where $q_i$ is a time invariant identity built from long horizon priors, and $r_i(t)$ is a short horizon, data driven refinement produced by diffusion conditioning during training.
>
> **Why this decomposition is reasonable.**  Additive modeling is a standard way to separate stable structure from fluctuating adjustments. This perspective appears in residual learning, where layers learn corrections around a baseline, and in classical time series decompositions that split a series into trend, seasonality, and remainder. **These precedents support using** **$q_i$** **to encode slowly varying identity and using** **$r_i(t)$** **to capture fast, context dependent deviations.**  In our design, long horizon priors built via multi cycle PCA and spectral embeddings preserve between node variance and aim to capture *inherent node properties* in $q_i$. The refinement $r_i(t)$ then *adapts these identities to current conditions* during training.
>
> This decomposition shifts the learning target from fitting a single all purpose node vector to correcting residuals under changing environments. A fully general theorem that connects this decomposition to end to end prediction error for all possible backbones is challenging and remains open. Our empirical results show consistent gains across architectures and datasets, which we offer as practical evidence, and we hope the presented formulation can serve as a basis for future theoretical analysis.
>
> ---
>
> > **W4. The topology priors in SNIP are constructed directly from a fixed adjacency matrix, which may not accurately capture the true correlations or dynamic dependencies among nodes. Prior studies in spatial-temporal forecasting have shown that such predefined graphs can be suboptimal or even misleading.**
>
> Thank you for raising this concern, we understand the point.
>
> To clarify, in SNIP **the topology prior** derived from a fixed adjacency **is not used in isolation**. We pair it with a time delayed interaction matrix that captures short term coupling, and ***we refine the concatenated priors*** during training with a small MLP and a diffusion style GCN. In this refinement, *the GCN’s adjacency serves as a local smoothing operator to adjust the prior in a short term, localized manner, rather than as a mechanism to hard encode or learn full spatio-temporal dependencies*. This design reduces the risk of over-trusting a possibly suboptimal predefined graph, while allowing data driven, short window signals to correct and adapt the prompts.

---

> ### Author Response · Authors · 2025-11-22
> **The Third Part of the Response to Reviewer MB12**
>
> > **Q1. Line 201, $X_\{:,:,1\}$ appears to be inconsistent with the earlier definition of X, which is a 3-dimensional tensor.**
>
> Thank you for your careful reading. We have corrected the notation and added context to avoid confusion: “For clarity, we describe the single-feature case ($C\=1$) below, which naturally extends to multi-channel inputs by concatenation.”
>
> ---
>
> > **Q2. The connection between the diffusion convolution operation and the prediction process during the expansion phase is not entirely clear. Is the diffusion convolution applied only during the stage of $\tau_2$ for training?**
>
> We are happy to provide further clarification. The diffusion graph convolution module serves as a prior refiner, so it is trained in both phases, pretraining on the base stage and fine tuning on the expansion stage. **At inference time it remains active to adjust the precomputed priors using the current input window, while its parameters are fixed.**
>
> End to end, SNIP has two parts, *offline prior construction*, and *an online adjustment block* that turns priors into effective node promptings conditioned on the short input sequence. This adjustment block, an MLP and a diffusion convolution, is trained jointly with the STF backbone and is executed at test time to produce the final prompts used by the predictor.
>
> We have clarified this in Section 4.2 and added stepwise algorithm pseudocode in Appendix A.3 to make the training and inference paths clear.
>
> ---
>
>
> > **Q3. Figure 1(b) is not very intuitive or informative in illustrating the differences among methods. The authors may consider redesigning it with clearer visual cues or a more concrete comparative example to better highlight the distinctions.**
>
> Thank you for the helpful suggestion. We have simplified and redesigned Figure 1(b) to make the contrasts among methods clearer and more concrete.
>
> ---
>
> Thank again for your valuable feedback. We will appropriately incorporate these detailed discussions into the final revised version of our paper. We appreciate your guidance!

---

> > ### Comment · Reviewer_MB12 · 2025-11-24
> >
> > I am overall satisfied with the response. Regarding W2, it seems that the model has a scalability issue (it takes several hours to process data scale of few hundred instances). The authors have acknowledged this limitation and a direction for future work. I will maintain my positive score and will refer to other reviewers in the discussion phase.

---

> > > ### Author Response · Authors · 2025-11-24
> > >
> > > Thank you for the follow up and for maintaining a positive score.
> > > A quick clarification on W2, the multi hour preprocessing does **NOT** refer to SNIP at the few hundred node scale. On EPeMS, **our SNIP’s offline prior construction takes 2.61 minutes** for all three priors on the same hardware and codebase, the multi hour figure refers to **GMAN**’s random walk based spatial embedding, about **252 minutes** in our profiling. Compared with methods that also rely on preprocessing, SNIP’s cost is acceptable and competitive, and the priors are cached for reuse across backbones and phases.

---

> > > ### Author Response · Authors · 2025-11-28
> > >
> > > Dear Reviewer MB12,
> > >
> > > Thank you for your careful review and constructive feedback. We hope our previous note has clarified the misunderstanding about computational cost. At the few hundred node scale on EPeMS, SNIP’s offline prior construction completes in **2.61 minutes**. Compared with other preprocessing based approaches, **SNIP’s cost is acceptable and competitive**.
> > >
> > > We believe this addresses your concern. **If it aligns with your assessment, we would be grateful if you could consider increasing the score**. Thank you again for your time and effort in reviewing our work. Your valuable comments helped us strengthen the manuscript.
> > >
> > > Best regards,
> > >
> > > The Authors

---

### Official Review · Reviewer_hAsY · 2025-11-01

**Soundness:** 3
**Presentation:** 4
**Contribution:** 3
**Rating:** 6
**Confidence:** 5

**Summary:**

This paper studies expanding-node spatial-temporal forecasting, a setting where sensor networks evolve over time: existing nodes may disappear, and new nodes with scarce history are added. Traditional spatio-temporal forecasting (STF) models assume a fixed node set and rely on node-specific learnable embeddings, which breaks in this setting because parameters scale with the number of nodes and new nodes have scarce data.

The authors propose SNIP (Structured Node Interaction Prompting), a model-agnostic prompting framework that decouples model parameters from node count. SNIP constructs static node priors from historical data along three axes: periodic priors obtained via PCA over repeated temporal cycles (e.g. daily, weekly) to capture node-specific long-term behavior; topology priors from spectral embeddings of the graph Laplacian to encode structural position; and time-delayed interaction priors from cross power spectral density (CSD), capturing lagged, asymmetric correlations between nodes and their dominant propagation delays.

These priors are concatenated and then dynamically refined during training via an MLP and diffusion graph convolution, yielding adaptive node promptings that can evolve over time. For new nodes, SNIP initializes priors using similarity-weighted mixtures of priors from "most similar" existing nodes (based on correlation strength), enabling few-shot adaptation; removed nodes incur no parameter cost because parameters are not tied to node identities.

The proposed approach named SNIPformer injects the refined promptings into an efficient spatio-temporal encoder. Across four datasets spanning traffic and renewable energy (EPeMS, PEMS04, SeaLoop, NREL-AL), and under simulated/base - expansion/test splits, SNIPformer outperforms baselines including node-agnostic forecasters (DLinear, iTransformer), modified STGNNs/Transformers without node embeddings (e.g. GWNET, STID, STAEformer), continual-learning style prompt-tuning (EAC, STKEC), and the recent expanding-variate baseline STEV.

**Strengths:**

- The paper is generally well written and well structured. It identifies and formalizes the expanding-node spatial-temporal forecasting problem: networks where nodes are added and removed between a base stage and an expansion stage, with very limited data for new nodes. This moves beyond the standard fixed-node assumption used by most traffic/energy forecasting models.

- The proposed method replaces learnable node embeddings with computed priors that encode periodic temporal signatures, graph topology, and lagged inter-node coupling. The idea of treating these priors as non-learnable node identity prompts and then refining them online is a novel take on node-specific conditioning without parameter-node coupling.

- Introduces a similarity-weighted initialization procedure for new nodes, which transfers priors from existing nodes in proportion to cross-correlation strength and dominant delay, and argues that this enables few-shot adaptation for nodes with almost no history.

- The experimental protocol is fairly thorough, with four datasets across traffic and energy domains, explicit base/expansion/test stages, and explicit partitioning of nodes into remain, deleted, and new.  The baselines span models without node-specific prompting (DLinear, iTransformer, DUETformer), STGNN / Transformer backbones with their node embeddings removed (GWNET, STID, STAEformer, etc.), continual learning / expanding-graph or prompt-tuning style methods (STKEC, EAC), and STEV, which is explicitly designed for expansion-like scenarios (Expanding-Variate TS Forecasting). SNIPformer achieves best or second-best MAE/RMSE in most settings, including for new nodes, which is the hardest regime.

- Ablation studies show that both static priors (especially periodic + inter-node priors) and dynamic refinement are necessary, and that higher-variance priors improve heterogeneity. The paper also argues that SNIP is more computationally efficient than STEV because it avoids full retraining after expansion and only requires lightweight fine-tuning with precomputed priors.

**Weaknesses:**

- Although the datasets are real traffic / energy datasets, most expansion scenarios (except EPeMS, which follows STEV) are synthetically constructed by partitioning nodes into remain / deleted / new groups and truncating history for new nodes. The paper does not evaluate on an actual incremental deployment log (e.g., "new sensors activated on these real dates, old sensors decommissioned here").

- The forecasting setup is 1-hour ahead with MAE/RMSE. There’s no multi-horizon stress test (longer-ahead prediction, where structural priors might degrade)

- Time-delayed interaction prior: depends on estimating pairwise cross-spectral structure using Welch’s method with a fixed window. The method assumes reasonably stationary local coupling. The paper does not quantify sensitivity to window size, noise, or nonstationary bursts (e.g., incidents).

**Questions:**

- The similarity-weighted initialization for new nodes relies on reliable cross-correlation with existing nodes, but when a node is truly new (short history, novel behavior, or located in a structurally new region), those similarities may be noisy or misleading. What if new nodes are structurally isolated?

- A related line you may wish to discuss is that of GAP-LSTM (which tackles spatio-temporal forecasting on geo-distributed sensor networks by explicitly modeling spatio-temporal autocorrelation via a hybrid of graph convolution, attention-augmented LSTMs, 2-D temporal convolutions, and latent memory states), DCRNN, and GMAN, which similarly couple graph operators with temporal sequence models/attention to capture directed diffusion and dynamic cross-node dependencies.

- For PEMS04 / SeaLoop / NREL-AL, new vs deleted nodes are sampled by partition, and new nodes’ histories are artificially truncated. Do you have (or could you collect) a real deployment timeline where sensors were physically added/removed over calendar time, and evaluate SNIP there? Even a partial case study would make the problem statement more concrete.

- STEV is your closest "expanding-variate" baseline. Can you clarify where SNIP’s gains come from in the datasets where you win?

- Could SNIP be layered onto very lightweight models (e.g. DLinear) and still give gains, or does it rely on having a spatial-temporal encoder downstream to actually use the prompts?

- See weaknesses

---

> ### Author Response · Authors · 2025-11-22
> **The First Part of the Response to Reviewer hAsY**
>
> **Response to Reviewer hAsY**
>
> Dear Reviewer hAsY,
>
> We sincerely thank you for taking the time and effort to provide valuable feedback on our paper. We have carefully considered each of your points and have addressed them one by one.
>
> ---
>
> > **W1: Although the datasets are real traffic / energy datasets, most expansion scenarios (except EPeMS, which follows STEV) are synthetically constructed by partitioning nodes into remain / deleted / new groups and truncating history for new nodes. The paper does not evaluate on an actual incremental deployment log (e.g., "new sensors activated on these real dates, old sensors decommissioned here").**
>
> We share your concern and appreciate the push toward truly incremental logs. At present, **public benchmarks that pair multivariate traffic or energy series with authoritative deployment or decommission timestamps are scarce**. Recent work that formalizes expanding variates, STEV, **explicitly notes the lack of publicly available EVTS data** and therefore crafts expansions by partitioning variables, using spatial anchors when locations are available, and random selection otherwise. These choices are documented in the STEV paper and its appendix.
>
> In our study, EPeMS follows the same _**area expansion**_ protocol in STEV, using location to emulate adding sensors in new spatial coverage. For the other datasets, although we have access to sensor locations or adjacency relationships, **they lack installation or retirement timestamps**. So we adopt the _**internal expansion**_ protocol by partitioning nodes into remain, deleted, and new, and by truncating the short post expansion history that is available for new nodes. This abstraction is consistent with STEV’s construction in which expansions are synthesized due to the absence of public EVTS logs.
>
> Finally, we simulate expansions under a practical premise, *sensors observe an underlying traffic or energy process, and adding or removing sensors does not alter that latent process.*  Under this premise, evaluating with spatially anchored or internally partitioned expansions, while withholding the short early histories of new nodes, is a reasonable proxy for changing node sets.
>
> We have revised the manuscript to state these assumptions explicitly, to label EPeMS as an area expansion case and the other datasets as internal expansion, and to reference the STEV protocol for clarity. We plan to collect datasets that reflect real deployment timelines in future work.
>
> ---
>
> > **W2: The forecasting setup is 1-hour ahead with MAE/RMSE. There’s no multi-horizon stress test (longer-ahead prediction, where structural priors might degrade)**
>
> Thank you for the suggestion. Long horizon performance is indeed a key concern. We have added multi horizon evaluations on EPeMS using a lightweight backbone, STID, with and without SNIP. We use historical 12, 24, 48, and 96 time steps to predict the subsequent 12, 24, 48, and 96 time steps, respectively.
>
>
> ||||12→12|||24→24|||48→48|||96→96||
> |------------|------------|------|-----:|-----|------|-----:|-----|------|-----:|-----|------|-----:|-----|
> |||All|Remain|New|All|Remain|New|All|Remain|New|All|Remain|New|
> |STID†|MAE|24.40|24.31|24.56|29.26|29.50|28.79|36.32|36.97|35.03|41.61|42.51|39.86|
> |STID†+SNIP|MAE|21.84|21.13|23.23|25.18|24.22|27.07|29.51|28.10|32.29|33.90|32.37|36.90|
> ||Improvements|10.49\%|13.11\%|5.40\%|13.94\%|17.91\%|5.97\%|18.73\%|24.01\%|7.83\%|18.54\%|23.85\%|7.43\%|
> |STID†|RMSE|37.38|37.44|37.23|44.22|44.85|42.92|53.83|55.14|51.14|59.92|61.59|56.49|
> |STID†+SNIP|RMSE|33.73|32.90|35.31|38.31|37.20|40.38|44.14|42.58|47.04|49.63|48.15|52.41|
> ||Improvements|9.75\%|12.14\%|5.16\%|13.38\%|17.07\%|5.92\%|18.01\%|22.78\%|8.02\%|17.16\%|21.81\%|7.21\%|
>
>
>
> Results show that **SNIP brings consistent gains across all horizons and node groups**. Relative improvements increase from 12 to 48 steps, with a modest attenuation at 96 steps. The trend indicates that SNIP is robust under multi step prediction.
>
> **Why this helps at longer horizons.**  The periodic prior aggregates stable long cycle statistics, for example daily and weekly components, which act as node specific identity prompts derived from richer historical context. **The dynamic refinement** then adjusts these prompts using short window time delayed interaction features, which ***mitigates potential drift of purely structural priors as the horizon grows.***
>
> We have added the full multi horizon results and analysis in the revised Appendix B.6.

---

> ### Author Response · Authors · 2025-11-22
> **The Second Part of the Response to Reviewer hAsY**
>
> > **W3: Time-delayed interaction prior: depends on estimating pairwise cross-spectral structure using Welch’s method with a fixed window. The method assumes reasonably stationary local coupling. The paper does not quantify sensitivity to window size, noise, or nonstationary bursts (e.g., incidents).**
>
> Thank you for raising this point. Below we first clarify our current configuration, then explain what this choice implies in the frequency and time domains, and finally report sensitivity on Hann window size and future plan.
>
> Our **current configuration** computes the cross spectral density (CSD) with Welch method using a Hann window, window length set to the forecasting horizon T \= 12, zero overlap. After obtaining CSD, we apply the inverse FFT to obtain the cross correlation sequence and take the peak location as the dominant delay and the peak magnitude as coupling strength. This is the standard cross spectrum to cross correlation link given by the Wiener–Khinchin relation for cross processes, and is the basis of generalized cross correlation style delay estimation.
>
> **Why this helps with noise.**  Welch’s averaging lowers variance of the spectral estimate by averaging modified periodograms across segments. The Hann window reduces leakage, which lowers the noise floor that would otherwise mask narrowband structure. Using frequency domain averaging before the IFFT therefore improves the robustness of the peak in the cross correlation sequence under broadband noise.
>
> **Sensitivity to window size.**  We quantified sensitivity to the Hann window length while keeping other settings fixed. Results are summarized below.
>
>
> |Hann window size|4|6|12|24|36|48|
> |----------------|----:|----:|----:|----:|----:|----:|
> |MAE|19.29|19.27|19.20|19.34|19.20|19.19|
> |RMSE|31.15|31.13|31.02|31.21|31.08|31.05|
>
>
> Performance varies within a narrow band and stabilizes as the window grows, which is consistent with Welch’s variance reduction and Hann’s leakage control. We set the window size to 12 in the main experiments as a balanced choice that matches the forecasting horizon and preserves time localization. We have added this sensitivity analysis in the Seciton 5.3 and Appendix B.3.
>
> **Scope and future work.**  We agree that a fuller study of the overlap setting, explicit noise level stress tests, and incident driven nonstationarity is important. We will extend the analysis along these axes in future work and include protocols to quantify robustness under those conditions.
>
> ---
>
> > **Q1:The similarity-weighted initialization for new nodes relies on reliable cross-correlation with existing nodes, but when a node is truly new (short history, novel behavior, or located in a structurally new region), those similarities may be noisy or misleading. What if new nodes are structurally isolated?**
>
> Thank you for the thoughtful question. It targets the edge case we also care about, namely a truly new node with short history, novel behavior, or placement in a previously uncovered region.
>
> **In our study, the similarity weighted initialization in SNIP is a transfer option, not a requirement.**  If a node lies in a structurally new region, we first compute its similarity to existing nodes using the short post-expansion history. Practitioners or researchers can set a reliability threshold. If the similarity falls below that threshold, we skip similarity-weighted aggregation and instead construct the three priors directly from the node’s own short history. We then fine tune on this short window so the model can adapt to the node’s unique pattern. As described in the *Appendix B.2*, **our empirical analysis recommends applying the similarity-weighted strategy only to the periodic prior for most datasets**, while not using it for the other two inter-node priors, since those capture more recent and volatile cross-node interactions.
>
> In the updated manuscript, we have rewritten the description and analysis of the similarity weighting strategy to prevent potential misunderstandings in Section 4.2 and Appendix B.2. Finally, distribution shift across datasets is an important concern, and we leave a fuller investigation of this issue to future work.

---

> ### Author Response · Authors · 2025-11-22
> **The Third Part of Response to Reviewer hAsY**
>
> > **Q2: A related line you may wish to discuss is that of GAP-LSTM (which tackles spatio-temporal forecasting on geo-distributed sensor networks by explicitly modeling spatio-temporal autocorrelation via a hybrid of graph convolution, attention-augmented LSTMs, 2-D temporal convolutions, and latent memory states), DCRNN, and GMAN, which similarly couple graph operators with temporal sequence models/attention to capture directed diffusion and dynamic cross-node dependencies.**
>
> Thank you for the pointer. These approaches primarily aim to model spatio-temporal correlations across nodes, for example by coupling graph operators with sequence models or attention to capture directed diffusion and dynamic cross-node dependencies. **Our focus is different yet complementary, we construct and refine node-specific identities that encode heterogeneity and relational signals.**
>
> Concretely, the cited methods are trained on short sliding windows, each window serves as a sample that emphasizes short-term spatio-temporal interactions. **In contrast, our framework first performs an offline analysis over all available history.**  From temporal and spatial dimensions we compute and average priors to form a node-specific identity prompt, then during training we apply dynamic refinement to correct any bias introduced by these priors under evolving conditions.
>
> The modeling objectives therefore emphasize different aspects. These models can be combined with our prompting framework (e.g., DCRNN and GMAN in Table 10 and Table 11, Appendix B.3), where **their sequence modeling modules capture short-term interactions and SNIP supplies stable, heterogeneous node identities that adapt online, yielding complementary benefits for spatio-temporal forecasting.**
> We have added these comparison in Related Work (Section 2).
>
> ---
>
> > **Q3: For PEMS04 / SeaLoop / NREL-AL, new vs deleted nodes are sampled by partition, and new nodes’ histories are artificially truncated. Do you have (or could you collect) a real deployment timeline where sensors were physically added/removed over calendar time, and evaluate SNIP there? Even a partial case study would make the problem statement more concrete.**
>
> Please see our response to W1 regarding the current availability of real deployment logs. In addition, to provide a concrete case style illustration, we visualize geographic distribution and performance comparison with three expanding scenarios (*Area Expansion*, *Spatial Expansion*, and _Internal Expansion_), in Appendix B.7. We are also initiating a broader effort to collect datasets that reflect real world expansion events and timelines. Thank you for the suggestion.

---

> ### Author Response · Authors · 2025-11-22
> **The Fourth Part of Response to Reviewer hAsY**
>
> > **Q4: STEV is your closest "expanding-variate" baseline. Can you clarify where SNIP’s gains come from in the datasets where you win?**
>
> Thank you for the question. The improvements primarily come from two sources.
>
> First, ***representation***. In STEV, node specific parameters are learned within short prediction windows, so each training sample reflects only *a limited temporal context*. Such windows can be insufficient to capture persistent node heterogeneity. SNIP constructs node identities as priors from *the full historical record*, *using PCA across multiple periodic scales*. This yields low dimensional, stable, and discriminative prompts that encode longer term behavior and preserve inter node differences, while being decoupled from the window length used during training.
>
> Second, ***adaptation***. SNIP adds a dynamic refinement module that treats the priors as a baseline and *learns corrections conditioned on the current short window and diffusion context*. This shifts the learning target from fitting an all purpose node vector to adjusting residuals under changing conditions. For example, when two nodes differ over the long run but become temporarily similar due to events such as congestion, the prior anchors the identity and the refinement adapts to the current regime. This reduces learning difficulty and improves robustness to regime shifts.
>
> Together, long horizon priors plus lightweight corrections lead to stronger generalization for new nodes with scarce history and greater stability for remain nodes, which explains the observed gains over STEV on the reported datasets.
>
> ---
>
> > **Q5: Could SNIP be layered onto very lightweight models (e.g. DLinear) and still give gains, or does it rely on having a spatial-temporal encoder downstream to actually use the prompts?**
>
>
> **Yes, SNIP can be layered onto very lightweight models and still give gains.**  We added six backbones in the generality study, including two lightweight MLP based models without explicit spatio temporal encoders, DLinear, which applies an MLP along the temporal dimension only, and STID, which applies an MLP along the feature dimension only. SNIP is injected as a node identity prompting module.
>
> The downstream backbone does not need graph layers, it only needs to accept the prompt vector at the input or embedding stage. SNIP performs its own lightweight refinement, so gains do not depend on a heavy spatio temporal encoder.
>
> Results on EPeMS and NREL-AL are shown below, with full multi dataset tables in the Appendix B.3 (Table 10 and Table 11).
>
>
> |||EPeMS|||NREL-AL||
> |-------|-----|------|-----|----|-------|----|
> ||All|Remain|New|All|Remain|New|
> |DLinear|32.70|32.26|33.56|2.54|2.59|2.41|
> |+AttP|32.37|31.91|33.25|2.50|2.55|2.37|
> |+SNIP|**29.13**|**28.95**|**29.47**|**2.19**|**2.23**|**2.08**|
> |STID†|24.40|24.31|24.56|2.00|2.03|1.89|
> |+AttP|24.35|24.26|24.53|2.01|2.05|1.90|
> |+SNIP|**21.84**|**21.13**|**23.23**|**1.86**|**1.89**|**1.77**|
>
>
> SNIP improves both DLinear and STID on All, Remain, and New groups. Notably, STID†+SNIP attains state of the art MAE on EPeMS among compared methods. This supports the claim that structured node prompting is beneficial.
>
> ---
>
> Thank you for your valuable feedback. We will appropriately incorporate these detailed discussions into the final revised version of our paper. Once again, we appreciate your suggestions!

---

> ### Comment · Reviewer_hAsY · 2025-11-27
>
> Thank you for the extensive work. I am satisfied with the response and would like to confirm my positive opinion of this manuscript. I will increase my score accordingly.

---

> > ### Author Response · Authors · 2025-11-28
> >
> > Dear Reviewer hAsY,
> >
> > Thank you for your thoughtful follow up and for confirming your positive assessment. We are grateful for your supportive evaluation and the increased score. We also sincerely appreciate your time and effort in reviewing our work. Your comments helped us clarify the presentation and strengthen the manuscript.
> >
> > Best regards,
> > The Authors

---

### Author Response · Authors · 2025-11-22
**General Response to All Reviewers**

We sincerely thank all the reviewers for their invaluable time and thoughtful feedback. **It is encouraging to see that reviewers have recognized the positive aspects of our manuscript.** Reviewer hAsY, MB12, and k2Rf highlighted `the importance and clear formalization` of the expanding node setting, and hAsY and MB12 noted the `novelty` of computed priors with online refinement plus the similarity weighted initialization for new nodes. All three observed `consistent gains`, with hAsY emphasizing `dataset and baseline breadth` and the `efficiency` from avoiding full retraining, and k2Rf underscoring the `static plus dynamic decomposition` and the `model agnostic design`.

We have provided point by point responses to all comments. Below is a concise summary of the major revisions incorporated into the new manuscript based on the reviewers’ suggestions. We hope these changes address the remaining concerns. (**Changes are marked in purple in the revised file.**)

- **Presentation**

    1. Based on the feedback from **Reviewer k2Rf**, we expanded Appendix A.3 with an end to end block diagram, and algorithm pseudocode for prior construction, prompting injection, diffusion refinement, and phase wise adaptation.
    2. Accroding to the feedback from **Reviewer MB12**, we clarified how diffusion convolution operates in pretraining, fine tuning, and inference in Section 4.2 and re-designed Figure 1(b) that contrasts method families with clear visual cues.

- **Experiments**

    1. Based on the feedback from **Reviewer k2Rf**:
        - we added generality experiment results on more diverse backbones (MLP based, graph based, attention based, and a hybrid MoE), with full tables in the appendix B.3 (Table 10 and Table 11).
        - Reported sensitivity studies for key hyperparameters, $k_\text{pca}$, $k_\text{topo}$, $k_\text{delay}$, $k_\text{corr}$, number of periodicities, and Hann window size for Welch CSD, in Section 5.3 and Appendix B.4.
        - Reported robustness to node ratio changes, varying new add and retire rates in Appendix B.3.
    2. Based on the feedback from **Reviewer MB12** and **k2Rf**: we added multi phase streaming evaluations on PEMS-Stream and Air-Stream, following the EAC protocol in Appendix B.5.
    3. Based on the feedback from **Reviewer hAsY**:
       - we added multi horizon tests on EPeMS, 12 to 96 steps, showing consistent gains, in Appendix B.6.
       - we added a visualization of prediction results for the two expansion modes (Area Expansion and Spatial Expansion) on the EPeMS dataset in Appendix B.7.

- **Clarifications and discussions**

    1. Accroding to the feedback from **Reviewer hAsY**: we stated the expansion protocol for each dataset, EPeMS as area expansion, others as internal expansion in Appendix B.2.
    2. Accroding to the feedback from **Reviewer MB12**:
       - Clarified similarity weighted initialization strategy, and documented the recommended policy to mix only the periodic prior in Section 4.2 and Appendix B.2.
       - Added a concise rationale for the Decomposition Hypothesis, static q plus dynamic r, with connections to additive modeling and residual style refinement in Section 4.2.


We sincerely appreciate the reviewers’ expertise, which has greatly strengthened our manuscript. We have made concrete revisions to address all raised issues and are grateful for the recognition and constructive suggestions. We respectfully welcome further discussion and guidance.

---

### Author Response · Authors · 2025-12-02
**Revision Summary and Reviewer Follow ups**

Dear Area Chair,

Thank you for your time and careful review. This note summarizes our revisions and the reviewers’ follow ups, so you can quickly assess the current state of the paper.

## Overall recognition from reviewers

- Reviewers **hAsY**, **MB12**, and **k2Rf** found `the problem setting important` and `clearly formalized`, expanding node STF with remain, deleted, and new nodes.

- Reviewers **hAsY** and **MB12** noted the `novelty` of using computed priors with online refinement and a similarity weighted initialization for new nodes with scarce history.

- All reviewers observed `consistent gains`, with **hAsY** emphasizing `dataset and baseline breadth` and the `efficiency` from avoiding full retraining, and **k2Rf** underscoring the `static plus dynamic decomposition` and the `model agnostic design`.

## Reviewer specific concerns and our revisions

- **Reviewer hAsY**, questioned the realism of our expansion setups, requested clearer multi horizon evidence, sensitivity around Welch windowing, and a sharper comparison to STEV, plus whether SNIP helps lightweight models.
  - We clarified the status of publicly available datasets and the expansion protocols, added visualizations illustrating the expansion modes, added multi horizon results on EPeMS, added Welch window sensitivity, further detailed the advantages compared to STEV, and reported gains on  lightweight backbones.

- **Reviewer MB12**, raised multi stage or streaming expansion, preprocessing cost at scale, theoretical footing for the static plus dynamic decomposition, fixed adjacency in topology priors, and several presentation issues.
  - We added multi stage results on the Stream datasets, quantified SNIP’s preprocessing cost and showed it is acceptable and competitive relative to other preprocessing based methods, provided a more concrete rationale for the static plus dynamic decomposition, reiterated that topology priors are further refined in the model, fixed presentation issues, and redesigned Figure 1(b).

- **Reviewer k2Rf**, requested validation on diverse backbones, larger and more dynamic streams, and comprehensive hyperparameter studies.
  - We extended the generality study to MLP based, graph based, attention based, and a hybrid MoE backbone, multi phase evaluations on PEMS-Stream and Air-Stream following EAC, sensitivity studies for key hyperparameters and Welch window size, and robustness under different new add and retire rates, expanded Appendix A.3 with an end to end diagram and pseudocode.

All changes are marked in purple in the revised file. More details can refer to *General Response to All Reviewers* and full responses to each reviewer.

## Reviewer follow ups and score changes

- **hAsY**, “*satisfied with the response*,” increased rating from 6 to 8 on Nov 28.

- **MB12**, “*satisfied with the response*,” kept rating 6. They initially misread our cost numbers, stating multi hour preprocessing for SNIP. However, our model’s preprocessing cost on EPeMS is 2.61 minutes, as stated in Section 5.4 of the original manuscript and the initial response, which is acceptable and competitive compared with other preprocessing based methods. We reiterated this clarification in the follow up.

- **k2Rf**, “*no further comments*,” raised the contribution subscore from 2 to 3 on Nov 26, kept rating 4.

In summary, the manuscript now has clearer methodology, stronger breadth and depth of experiments, explicit sensitivity and robustness analyses. We believe the remaining concerns have been addressed and the overall quality has improved.

We sincerely appreciate the reviewers’ expertise, which has strengthened our manuscript. Thank you again for your careful effort and consideration.

Best regards,

The Authors

---

### Meta-Review · Area_Chair_aJR1 · 2026-01-06

**Summary:**

This paper proposes structured node interaction prompting for expanding-node spatial-temporal forecasting. I have carefully read the revised manuscript and, the reviews, the rebuttals, and the responses from each reviewer. I appreciate the tremendous efforts by the authors in addressing the reviewers’ comments. However, several fundamental concerns remain unresolved, which make it difficult for me to recommend acceptance.

First, I agree with all reviewers that the proposed expanding-node setting is largely synthetic and may not carry sufficient practical significance in real-world deployments.
The experimental settings/datasets are largely inherited from STEV, and the proposed method is mainly validated based on Table 1. This substantially limits the credibility and generality of the empirical claims. For the experiments, this manuscript should be more self-contained. Now the audience have to refer to STEV for more details.

Second, this paper is not so well contextualized within the existing related works. The novelty of this paper should be better differentiated from the existing studies on spatiotemporal ood learning or spatiotemporal transfer learning. Many related works on spatiotemporal ood or spatiotemporal transfer learning should be added as comparison baselines to demonstrate the real advantage of the proposed method. Besides, in my opinion, we don’t have to create a new research problem called expanding-node spatiotemporal forecasting.

Third, despite additional experiments, questions remain regarding scalability and robustness.

**Reviewer Concerns:**

Some experimental details are clarified by the authors in the revision and responses, while the core concerns listed above still remain.

**Reviewer Scores:**

Reviewer hAsY, increased rating from 6 to 8.

Reviewer MB12, kept rating 6.

Reviewer k2Rf, kept rating 4.

---

### Decision · Program_Chairs · 2026-01-26

Reject